# A Size-Dependent Finite Element Method for the 3D Free Vibration Analysis of Functionally Graded Graphene Platelets-Reinforced Composite Cylindrical Microshells Based on the Consistent Couple Stress Theory

**DOI:** 10.3390/ma16062363

**Published:** 2023-03-15

**Authors:** Chih-Ping Wu, Tech-Fatt Tan, Hao-Ting Hsu

**Affiliations:** Department of Civil Engineering, National Cheng Kung University, Tainan City 701, Taiwan

**Keywords:** consistent couple stress theory, finite element method, free vibration, functionally graded cylindrical shells, graphene platelets, 3D analysis

## Abstract

Within a framework of the consistent couple stress theory (CCST), a size-dependent finite element method (FEM) is developed. The three-dimensional (3D) free vibration characteristics of simply-supported, functionally graded (FG) graphene platelets (GPLs)-reinforced composite (GPLRC) cylindrical microshells are analyzed. In the formulation, the microshells are artificially divided into numerous finite microlayers. Fourier functions and Hermitian *C*^2^ polynomials are used to interpolate the in-surface and out-of-surface variations in the displacement components induced in each microlayer. As a result, the second-order derivative continuity conditions for the displacement components at each nodal surface are satisfied. Five distribution patterns of GPLs varying in the thickness direction are considered, including uniform distribution (UD) and FG A-type, O-type, V-type, and X-type distributions. The accuracy and convergence of the CCST-based FEM are validated by comparing the solutions it produces with the exact and approximate 3D solutions for FG cylindrical macroshells reported in the literature, for which the material length scale parameter is set at zero. Numerical results show that by increasing the weight fraction of GPLs by 1%, the natural frequency of FG-GPLRC cylindrical microshells can be increased to more than twice that of the homogeneous cylindrical microshells. In addition, the effects of the material length scale parameter, the GPL distribution patterns, and the length–to–thickness ratio of GPLs on natural frequencies of the FG-GPLRC cylindrical microshells are significant.

## 1. Introduction

In recent decades, due to the successive discovery of nanoscale materials, namely, carbon nanotubes (CNTs) and graphene sheets (GSs), the development of material technology has undergone rapid changes [1,2]. Due to the superior physical, chemical, thermal, and electrical material properties of CNTs and GSs, the application of CNTs and GSs has gradually become popular in cutting-edge technology industries, such as micro-electro-mechanical systems and nano-electro-mechanical systems, sensors and actuators, aerospace, submarine, automobile, and biological technologies [3,4,5]. On the one hand, because of their extreme properties of high stiffness–to–weight ratio and high strength–to–weight ratio, CNTs and GPLs have been used as reinforcement materials and have been embedded in specific matrices to form nanocomposite structures, with some predesigned functions of CNT and GPL distribution patterns varying in the thickness direction [6,7,8]. On the other hand, various systems and components used in the above-mentioned cutting-edge technologies have been gradually miniaturized. Therefore, the relevant mechanical analyses of CNT-reinforced composite (CNTRC) and GPLRC microstructures have attracted considerable attention.

It has been well-known that due to the size-dependent effect, the classical continuum mechanics is no longer applicable to various mechanical analyses of microstructures [9,10,11]. In order to capture the size-dependent effect, some higher-order nonclassical continuum mechanics-based theories have been developed in the past several decades, including the nonlocal elasticity theory [12,13,14], the micropolar elasticity [15], doublet mechanics [16], the strain gradient theory [17], and the couple stress theory (CST) [18]. This article reviews the development of the CST, the CST-based classical shell theories, and the CST-based shear deformation shell theories.

Due to Eringen’s criticism [19] of the uncertainty problem of the CST, Yang et al. [20] and Hadjesfandiari and Dargush [21,22] overcame these issues with the CST. They successively developed the modified CST (MCST) and the CCST based on a symmetric and a skew-symmetric couple stress tensor assumption, respectively. Afterward, the material length scale coefficients required for analyzing elastic isotropic solids were reduced from two to one, making the MCST and the CCST practically feasible. The MCST and CCST have been successfully applied to analyze the torsional behavior of a thin cylinder and the pure bending behavior of a flat plate with an infinite width [20,21].

Based on the MCST, some two-dimensional (2D) microstructure-dependent shell theories have been developed for analyzing various mechanical behaviors of FG cylindrical and conical microshells. Tang et al. [23] developed a microstructure-dependent Kirchhoff thin plate theory to study the microplate’s static bending, buckling, and free vibration behaviors. Incorporating the kinematic model of Love’s classical shell theory (LCST) into the MCST, Mehralian and Beni [24] developed a size-dependent LCST for the buckling analysis of single-walled CNTs (SWCNTs). The SWCNTs of interest were placed on a Winkler-Pasternak foundation and were subjected to axial compression, electric voltages, and constant temperature changes. Afterward, based on the MCST-based LCST, Zeighampour and Beni [25] examined the free vibration characteristics of a single-walled carbon conical nanoshell. The results showed that the material length scale parameter causes the nanoshell to become stiffer, increasing the nanoshell’s lowest natural frequencies. Liu and Wang [26] presented size-dependent free vibration and buckling analyses of 3D graphene foam cylindrical microshells. In conjunction with the MCST and the kinematic model of the LCST, Soleimani et al. [27] developed the weak formulation of a two-node size-dependent axisymmetric shell element. Then, they applied it to carry out a buckling analysis of circular microplates, conical microshells, and cylindrical microshells. Combining the MCST with the kinematic model of the first-order shear deformation theory (FSDT), Gholami et al. [28] analyzed the axial buckling and dynamic instability behaviors of FG cylindrical microshells. The corresponding governing equations were derived using Hamilton’s principle and were written as Mathieu-Hill equations. Bolotin’s method was used to determine the instability regions. Afterward, with this MCST-based FSDT, Salehipour et al. [29] studied the size-dependent free vibration and static bending behaviors of FG porous microshells and nanoshells under clamped and simply-supported boundary conditions; Zeighampour and Beni [30] examined the free vibration characteristics of SWCNTs. Esfahani et al. [31] analyzed the vibration characteristics of a conical sandwich shell with a saturated FG porous core. Effective material properties of the porous shell were estimated using Biot’s theory, and the deformations of the porous shell were described using the FSDT. Shameli et al. [32] studied nanorods’ free torsional vibration characteristics with noncircular cross-sections based on the second-order strain gradient theory. Incorporating the displacement model of the higher-order shear deformation theory into the MCST, Lyu et al. [33] investigated the thermo-electro-mechanical free vibration and buckling behaviors of an FG piezoelectric porous cylindrical microshell. Based on the modified strain gradient theory, Thai et al. [34] presented an isogeometric analysis (IGA) for studying the postbuckling behavior of an FG microplate subjected to mechanical and thermal loads. In their formulation, the kinematic model of Reddy’s refined shear deformation theory (RSDT) and the von Kármán geometrical nonlinearity (VKGN) were employed to describe the deformation of the microplate. The nonuniform rational B-splines basis functions were used to generate the corresponding *C*^2^ interpolation (shape) functions for each primary variable. The symbol *C*^2^ represents the continuity conditions of the order of the derivatives for each primary variable up to and including the second order. Afterward, Thai et al. [35] employed this IGA approach to study the nonlinear static bending and the nonlinear transient vibration behaviors of FG microplates. Nguyen et al. [36] used this IGA approach to analyze the vibration characteristics of cracked FG microplates.

The kinematic models of the LCST, FSDT, RSDT, and sinusoidal shear deformation theory (SSDT) have also been incorporated into the MCST for analyzing various mechanical behaviors of FG-GPLRC microshells. Wang et al. [37] used the MCST-based LCST to analyze the size-dependent vibration of FG-GPLRC cylindrical microshells. In the analysis, they considered four GPL distribution patterns. They estimated the effective Young’s modulus and Poisson’s ratio using the modified Halpin-Tsai model [38] and the rule of mixtures [38], respectively. Baghbadorani and Kiani [39] presented a size-dependent analysis for the free vibration characteristics of FG-GPLRC cylindrical microshells using the MCST-based FSDT. With the MCST-based FSDT, Safarpour et al. [40] analyzed the thermal buckling and the free and forced vibration behavior of FG multilayered GPLRC nanostructures resting on a Winkler-Pasternak foundation; Salehi et al. [41] carried out the nonlinear vibration analysis of an imperfect FG-GPLRC porous cylindrical nanoshell. Wang et al. [42] conducted size-dependent research for the dynamic instability behavior of FG-GPLRC cylindrical micropanels using the MCST-based SSDT. Based on the MCST, Moayedi et al. [43] reformulated the classical continuum mechanics-based RSDT to study the size-dependent thermal buckling responses of FG-GPLRC micropanels.

After a close literature survey, we found that almost all relevant mechanical analyses of FG cylindrical microshells involved using size-dependent 2D theories, not the size-dependent 3D approach for elastic solids. In order to completely capture the 3D effects on the structural behavior of an FG microplate, including thickness stretching effects, shear deformation effects, 3D couple-stress tensor effects, and zig-zag deformation effects (for laminated microscale plates), Wu and Hsu [44] and Wu and Lu [45] developed the Lagrangian *C*^0^ and the Hermitian *C*^1^ FEMs based on the CCST to analyze FG elastic microplates. By specifying the material length scale parameter as zero, their results agree well with the exact and approximate 3D solutions of FG macroplates reported in the literature. The results also showed that the convergence rate of the Hermitian *C*^1^ FEM was more rapid than that of the Lagrangian *C*^0^ FEM. In order to further speed up the convergence rate of the Hermitian *C*^1^ FEMs and extend their application from the analysis of microplates to the study of cylindrical shells, we aim to develop a Hermitian *C*^2^ FEM based on the CCST to investigate the 3D free vibration characteristics of a simply-supported FG-GPLRC cylindrical microshell. A 3D weak formulation to address the current issue is first derived using Hamilton’s principle by selecting the displacement components as the primary variables. Then, Fourier functions and Hermitian *C*^2^ polynomials are used to interpolate the variations of the primary variables in the nodal surface and the thickness direction, respectively. As a result, the second-order derivative continuity conditions for the primary variables at each nodal surface are satisfied. In the numerical examples, five GPL distribution patterns varying in the thickness direction are considered, with a constant weight fraction of GPLs, including UD and FG A-type, O-type, V-type, and X-type distributions. A parametric study related to some key effects on the lowest natural frequencies of the FG-GPLRC cylindrical microshell is conducted, including the impacts of the material length scale parameter, the GPL distribution patterns, the GPL weight fractions, the length–to–mid-surface radius and mid-surface radius–to–thickness ratios of the microshell, and the length–to–thickness ratio of GPLs.

## 2. Effective Material Properties

The schematic diagram of a simply-supported FG cylindrical microshell of interest in this study is shown in Figure 1, for which the dimensions of the microshell fall within 0≤x≤L, 0≤θ≤2π, and a≤r≤b. The variables *x*, θ, and *r* are the cylindrical shell coordinates. The variables *a* and *b* denote the inner and outer radii of the microshell, respectively. The variables *h*, *L*, and *R* represent the thickness, the length, and the mid-surface radius of the microshell, respectively. Finally, the variable ζ denotes the global thickness coordinate of the microshell measured from the mid-surface. Some relationships between the above geometric variables are given as r=R+ζ, *a* = *R* − (*h*/2), and *b* = *R* + (*h*/2).

The microshell of interest is made of FG-GPLRC material, which is formed by mixing a specific matrix material and a reinforcement material GPL according to a particular function of GPL distribution pattern, varying in the thickness direction when the weight fraction ($W_{GPL}^{*}$) of GPLs is fixed. Five GPL distribution patterns varying in the thickness direction are considered in the following numerical examples: UD and FG A-type, O-type, V-type, and X-type distributions. In addition, the Halpin-Tsai model [36] is used to estimate the effective Young’s modulus, and the rule of mixtures [36] is used to estimate the effective Poisson’s ratio and the effective mass density. Finally, the following equations are used to describe these material properties.

According to the Halpin-Tsai model, the effective Young’s modulus of the FG-GPLRC cylindrical microshell $E_{eff}$ can be approximated with
(1)$$E_{eff}=\left(3/8\right)E_{L}+\left(5/8\right)E_{T},$$
where $E_{L}$ and $E_{T}$ denote the longitudinal and transverse moduli, respectively, which are expressed following Yang et al. [46] as:(2)$$E_{L}=\left[\left(1+\xi_{L}\eta_{L}V_{GPL}\right)/\left(1-\eta_{L}V_{GPL}\right)\right]E_{matrix},$$
(3)$$E_{T}=\left[\left(1+\xi_{T}\eta_{T}V_{GPL}\right)/\left(1-\eta_{T}V_{GPL}\right)\right]E_{matrix},$$
(4)$$\eta_{L}=\left[\left(E_{GPL}/E_{matrix}\right)-1\right]/\left[\left(E_{GPL}/E_{matrix}\right)+\xi_{L}\right],$$
(5)$$\eta_{T}=\left[\left(E_{GPL}/E_{matrix}\right)-1\right]/\left[\left(E_{GPL}/E_{matrix}\right)+\xi_{T}\right],$$
(6)$$\xi_{L}=2\left[{\left(L_{x}\right)}_{GPL}/h_{GPL}\right],$$
(7)$$\xi_{T}=2\left[{\left(L_{y}\right)}_{GPL}/h_{GPL}\right],$$
where the variables $E_{GPL}\;\mathrm{and}\;E_{matrix}$ denote Young’s modulus of GPLs and matrix material, respectively; the variable $V_{GPL}$ is the volume fraction of GPLs; the variables $\xi_{L}\;\mathrm{and}\;\xi_{T}$ are the parameters characterizing the geometrical dimensions of GPLs; the variables $\eta_{L}\;\mathrm{and}\;\eta_{T}$ are the parameters describing the geometrical dimensions of GPLs and Young’s modulus ratio between GPLs and matrix material; the variables ${\left(L_{x}\right)}_{GPL},$ ${\left(L_{y}\right)}_{GPL},$ and $h_{GPL}$ denote the length, width, and thickness of GPLs, respectively.

According to the rule of mixtures, the effective Poisson’s ratio $\upsilon_{eff}$ and the effective mass density $\rho_{eff}$ of the GPLRC material are given as
(8)$$\upsilon_{eff}=\upsilon_{matrix}+\left(\upsilon_{GPL}-\upsilon_{matrix}\right)V_{GPL},$$
(9)$$\rho_{eff}=\rho_{matrix}+\left(\rho_{GPL}-\rho_{matrix}\right)V_{GPL},$$
where subscripts *GPL*, *matrix*, and *eff* denote GPLs, matrix material, and GPLRC material, respectively. The variables $V_{GPL}\;\mathrm{and}\;V_{matrix}$ are the volume fractions of GPLs and matrix material, respectively. In the numerical example, with a specific value of $W_{GPL}^{*}$, the weight fractions of five relevant distribution patterns of GPLs varying in the thickness direction of the microshell ($W_{GPL}\left(\zeta \right)$) are expressed as follows:(10)$$W_{GPL}\left(\zeta \right)=W_{GPL}^{*},\;\;\;\;(\mathrm{for}\;\mathrm{the}\;\mathrm{UD}),$$
(11)$$W_{GPL}\left(\zeta \right)=\left[1-\left(2\zeta /h\right)\right]W_{GPL}^{*},\;\;(\mathrm{for}\;\mathrm{the}\;\mathrm{FG}\;A\text{-}\mathrm{type}\;\mathrm{distribution}),$$
(12)$$W_{GPL}\left(\zeta \right)=\left[1-\left(2\left|\zeta \right|/h\right)\right]W_{GPL}^{*},\;\;(\mathrm{for}\;\mathrm{the}\;\mathrm{FG}\;O\text{-}\mathrm{type}\;\mathrm{distribution}),$$
(13)$$W_{GPL}\left(\zeta \right)=\left[1+\left(2\zeta /h\right)\right]W_{GPL}^{*},\;\;(\mathrm{for}\;\mathrm{the}\;\mathrm{FG}\;V\text{-}\mathrm{type}\;\mathrm{distribution}),$$
(14)$$W_{GPL}\left(\zeta \right)=2\left(2\left|\zeta \right|/h\right)W_{GPL}^{*},\;\;\;(\mathrm{for}\;\mathrm{the}\;\mathrm{FG}\;X\text{-}\mathrm{type}\;\mathrm{distribution}),$$
where the weight fractions of the above five GPL distribution patterns varying in the thickness direction are shown in Figure 2. The total volume fraction of GPLs in each case in Equations (10)–(14) can be obtained using the relationship between $V_{GPL}$ and $W_{GPL}$, which is $V_{GPL}=W_{GPL}/\left[W_{GPL}+\left(\rho_{GPL}/\rho_{matrix}\right)\left(1-W_{GPL}\right)\right],$

## 3. The 3D CCST for an Elastic Solid

Within a framework of the CCST for an elastic solid developed by Hadjesfandiari and Dargush [21,22], the force-stress tensor ($\sigma_{ij}$) at a material point induced in a deformed elastic solid is assumed to be asymmetric, and the couple-stress tensor ($\mu_{ij}$) is skew-symmetric. Hadjesfandiari and Dargush thus decomposed the force-stress tensor into the symmetric part ($\sigma_{\left(ij\right)}$) and the skew-symmetric part ($\sigma_{\left[ij\right]}$), with the former and the latter being distinguished using parentheses and brackets surrounding a pair of indices, respectively. In the formulation presented by Hadjesfandiari and Dargush, the skew-symmetric part of the force-stress tensor ($\sigma_{\left[ij\right]}$) was expressed in terms of the couple-stress tensor ($\mu_{ij}$) as follows:(15)$$\sigma_{\left[ji\right]}=-\left(1/2\right)\left(\mu_{i,j}-\mu_{j,i}\right),$$
where subscripts *i*, *j*, and *k* permute in a natural order; and $\mu_{k}=\mu_{ji}=-\mu_{ij}$.

The strain energy density of an elastic microsolid is a function of the strain tensor ($\varepsilon_{ij}$) and the skew-symmetric part of the curvature tensor ($\kappa_{ij}$). The strain tensor is a symmetric tensor conjugated with the symmetric part of the force-stress tensor ($\sigma_{\left(ij\right)}$), and the skew-symmetric part of the curvature tensor ($\kappa_{ij}$) is conjugated with the couple-stress tensor ($\mu_{ij}$), which is a skew-symmetric tensor. Therefore, the strain energy in a microscale elastic solid occupying a volume Ω can be written as follows:(16)$$U_{s}=\int_{\Omega }\left[\left(1/2\right)\sigma_{\left(ij\right)}\varepsilon_{ij}-\mu_{ij}\kappa_{ij}\right]d\Omega ,$$
where $\sigma_{\left(ij\right)}=C_{ijkl}\varepsilon_{kl}$, and *C_ijkl_* is the elastic coefficient; $\varepsilon_{kl}=\left(u_{k,l}+u_{l,k}\right)/2$; $\kappa_{ij}=\left(\theta_{i,j}-\theta_{j,i}\right)/2$, $\theta_{i}=\theta_{kj}=\left(u_{k,j}-u_{j,k}\right)/2$, and $\theta_{kj}$ denotes the rotation tensor, which is a skew-symmetric tensor; and $\mu_{ij}=-8Gl^{2}\kappa_{ij}$ for an elastic solid, where *G* is the shear modulus and *l* denotes the material length scale parameter characterizing the size-dependent effects. The detailed expressions for the relationships between the above tensors and the displacement comments are given in the following section.

## 4. The CCST-Based Semi-Analytical Finite Element Formulation

### 4.1. Kinematics Assumptions

Based on the CCST for an elastic solid, we develop a weak formulation for analyzing the free vibration characteristics of a simply-supported FG-GPLRC cylindrical microshell. In the formulation, the microshell is artificially divided into *n_l_* cylindrical microlayers, for which the domain of each microlayer falls within $0\leq x\leq L_{x}$, 0≤θ≤2π, $\left(-h_{m}/2\right)\leq z_{m}\leq \left(h_{m}/2\right)$, and *m* = 1 − *n_l_*. The variable $z_{m}$ represents the local thickness coordinate measured from the midsurface of each microlayer. The elastic displacement components of each microlayer of the FG-GPLRC cylindrical microshell for each Hermitian *C*^2^ element are given as follows:(17)$$\begin{array}{ll} u_{x}^{\left(m\right)}\left(x,\theta ,z_{m},t\right) & =\sum_{i=1}^{n_{d}}\left[\psi_{3i-2}^{\left(m\right)}\left(z_{m}\right)u_{i}^{\left(m\right)}\left(x,\theta ,t\right)+\psi_{3i-1}^{\left(m\right)}\left(z_{m}\right)\theta_{ui}^{\left(m\right)}\left(x,\theta ,t\right)+\psi_{3i}^{\left(m\right)}\left(z_{m}\right)\kappa_{ui}^{\left(m\right)}\left(x,\theta ,t\right)\right]\\ & =\sum_{i=1}^{n_{d}}\psi_{i}^{\left(m\right)}d_{ui}^{\left(m\right)}, \end{array}$$
(18)$$\begin{array}{ll} u_{\theta }^{\left(m\right)}\left(x,\theta ,z_{m},t\right) & =\sum_{i=1}^{n_{d}}\left[\psi_{3i-2}^{\left(m\right)}\left(z_{m}\right)v_{i}^{\left(m\right)}\left(x,\theta ,t\right)+\psi_{3i-1}^{\left(m\right)}\left(z_{m}\right)\theta_{vi}^{\left(m\right)}\left(x,\theta ,t\right)+\psi_{3i}^{\left(m\right)}\left(z_{m}\right)\kappa_{vi}^{\left(m\right)}\left(x,\theta ,t\right)\right]\\ & =\sum_{i=1}^{n_{d}}\psi_{i}^{\left(m\right)}d_{vi}^{\left(m\right)}, \end{array}$$
(19)$$\begin{array}{ll} u_{r}^{\left(m\right)}\left(x,\theta ,z_{m},t\right) & =\sum_{i=1}^{n_{d}}\left[\psi_{3i-2}^{\left(m\right)}\left(z_{m}\right)w_{i}^{\left(m\right)}\left(x,\theta ,t\right)+\psi_{3i-1}^{\left(m\right)}\left(z_{m}\right)\theta_{wi}^{\left(m\right)}\left(x,\theta ,t\right)+\psi_{3i}^{\left(m\right)}\left(z_{m}\right)\kappa_{wi}^{\left(m\right)}\left(x,\theta ,t\right)\right]\\ & =\sum_{i=1}^{n_{d}}\psi_{i}^{\left(m\right)}d_{wi}^{\left(m\right)}, \end{array}$$
where *t* denotes the time variable; *n_d_* represents the number of nodal surfaces in each microlayer; the superscript *m* indicates the *m*th-microlayer; the variables $u_{i}^{\left(m\right)}$, $v_{i}^{\left(m\right)}$, and $w_{i}^{\left(m\right)}$ are the elastic displacement components in *x*, θ, and *r* axes, respectively, on the *i*th-nodal surface of the *m*th-microlayer of the FG-GPLRC cylindrical microshell. The variables $\theta_{ui}^{\left(m\right)}$, $\theta_{vi}^{\left(m\right)}$, and $\theta_{wi}^{\left(m\right)}$ are the first-order derivatives of $u_{i}^{\left(m\right)}$, $v_{i}^{\left(m\right)}$, and $w_{i}^{\left(m\right)}$ with respect to the thickness coordinate, respectively. The variables $\kappa_{ui}^{\left(m\right)}$, $\kappa_{vi}^{\left(m\right)}$, and $\kappa_{wi}^{\left(m\right)}$ are the second-order derivatives of $u_{i}^{\left(m\right)}$, $v_{i}^{\left(m\right)}$, and $w_{i}^{\left(m\right)}$ with respect to the thickness coordinate, respectively. The variables $\psi_{i}^{\left(m\right)}$ ($i=1,2,\ldots ,3n_{d}$) are the shape (or interpolation) functions that consist of Hermitian *C*^2^ polynomial functions and satisfy the continuity conditions for the second-order derivatives of the primary variables at each nodal surface; and $\psi_{i}^{\left(m\right)}=\left\{\begin{array}{cc} \psi_{3i-2}^{\left(m\right)} & \psi_{3i-1}^{\left(m\right)}\;\psi_{3i}^{\left(m\right)} \end{array}\right\},$
(20)$$d_{ui}^{\left(m\right)}=\left\{\begin{array}{c} u_{i}^{\left(m\right)}\\ \begin{array}{l} \theta_{ui}^{\left(m\right)}\\ \kappa_{ui}^{\left(m\right)} \end{array} \end{array}\right\},\;d_{vi}^{\left(m\right)}=\left\{\begin{array}{c} v_{i}^{\left(m\right)}\\ \begin{array}{l} \theta_{vi}^{\left(m\right)}\\ \kappa_{vi}^{\left(m\right)} \end{array} \end{array}\right\},\;d_{wi}^{\left(m\right)}=\left\{\begin{array}{c} w_{i}^{\left(m\right)}\\ \begin{array}{l} \theta_{wi}^{\left(m\right)}\\ \kappa_{wi}^{\left(m\right)} \end{array} \end{array}\right\},\;\mathrm{and}\;d_{\phi i}^{\left(m\right)}=\left\{\begin{array}{c} \phi_{i}^{\left(m\right)}\\ \begin{array}{l} \theta_{\phi i}^{\left(m\right)}\\ \kappa_{\phi i}^{\left(m\right)} \end{array} \end{array}\right\}.$$

For each microlayer, the linear constitutive equations, which are valid for the orthotropic elastic materials, following Hadjesfandiari and Dargush [21,22], are given as follows:(21)$$\left\{\begin{array}{c} \sigma_{\left(xx\right)}^{\left(m\right)}\\ \sigma_{\left(\theta \theta \right)}^{\left(m\right)}\\ \sigma_{\left(rr\right)}^{\left(m\right)}\\ \sigma_{\left(\theta r\right)}^{\left(m\right)}\\ \sigma_{\left(xr\right)}^{\left(m\right)}\\ \sigma_{\left(x\theta \right)}^{\left(m\right)} \end{array}\right\}=\left[\begin{array}{cccccc} c_{11}^{\left(m\right)} & c_{12}^{\left(m\right)} & c_{13}^{\left(m\right)} & 0 & 0 & 0\\ c_{12}^{\left(m\right)} & c_{22}^{\left(m\right)} & c_{23}^{\left(m\right)} & 0 & 0 & 0\\ c_{13}^{\left(m\right)} & c_{23}^{\left(m\right)} & c_{33}^{\left(m\right)} & 0 & 0 & 0\\ 0 & 0 & 0 & c_{44}^{\left(m\right)} & 0 & 0\\ 0 & 0 & 0 & 0 & c_{55}^{\left(m\right)} & 0\\ 0 & 0 & 0 & 0 & 0 & c_{66}^{\left(m\right)} \end{array}\right]\left\{\begin{array}{c} \varepsilon_{xx}^{\left(m\right)}\\ \varepsilon_{\theta \theta }^{\left(m\right)}\\ \varepsilon_{rr}^{\left(m\right)}\\ \gamma_{\theta r}^{\left(m\right)}\\ \gamma_{xr}^{\left(m\right)}\\ \gamma_{x\theta }^{\left(m\right)} \end{array}\right\},$$
(22)$$\left\{\begin{array}{c} \mu_{x}^{\left(m\right)}\\ \mu_{\theta }^{\left(m\right)}\\ \mu_{r}^{\left(m\right)} \end{array}\right\}=-\left(\frac{1}{2}\right)\left[\begin{array}{ccc} b_{11}^{\left(m\right)} & 0 & 0\\ 0 & b_{22}^{\left(m\right)} & 0\\ 0 & 0 & b_{33}^{\left(m\right)} \end{array}\right]\left\{\begin{array}{c} \kappa_{x}^{\left(m\right)}\\ \kappa_{\theta }^{\left(m\right)}\\ \kappa_{r}^{\left(m\right)} \end{array}\right\},$$
where the variables $\sigma_{\left(xx\right)}^{\left(m\right)},\;\sigma_{\left(\theta \theta \right)}^{\left(m\right)},\;\sigma_{\left(rr\right)}^{\left(m\right)},\;\sigma_{\left(\theta r\right)}^{\left(m\right)},\;\sigma_{\left(xr\right)}^{\left(m\right)},\;\mathrm{and}\;\sigma_{\left(x\theta \right)}^{\left(m\right)}$ represent the symmetric part of the force-stress components. The variables $\mu_{x}^{\left(m\right)},\;\mu_{\theta }^{\left(m\right)},\;\mathrm{and}\;\mu_{r}^{\left(m\right)}$ are the couple-stress components, for which $\mu_{x}^{\left(m\right)}=\mu_{r\theta }^{\left(m\right)}=-\mu_{\theta r}^{\left(m\right)},$ $\mu_{\theta }^{\left(m\right)}=\mu_{xr}^{\left(m\right)}=-\mu_{rx}^{\left(m\right)},$ $\mathrm{and}\;\mu_{r}^{\left(m\right)}=\mu_{\theta x}^{\left(m\right)}=-\mu_{x\theta }^{\left(m\right)};$ the variables $\varepsilon_{xx}^{\left(m\right)},\;\varepsilon_{\theta \theta }^{\left(m\right)},\;\varepsilon_{rr}^{\left(m\right)},$ $\gamma_{\theta r}^{\left(m\right)},\;\gamma_{xr}^{\left(m\right)},$ $\mathrm{and}\;\gamma_{x\theta }^{\left(m\right)}$ are the strain components. The variables $\kappa_{x}^{\left(m\right)},\;\kappa_{\theta }^{\left(m\right)},\;\mathrm{and}\;\kappa_{r}^{\left(m\right)}$ represent the skew-symmetric part of the curvature tensor. The variables $c_{ij}^{\left(m\right)}$ and $b_{kk}^{\left(m\right)}$ denote the elastic coefficients and the material length scale coefficients, respectively, where $b_{11}^{\left(m\right)}=16G_{r\theta }^{\left(m\right)}l_{1}^{2},$ $b_{22}^{\left(m\right)}=16G_{xr}^{\left(m\right)}l_{2}^{2},$ $b_{33}^{\left(m\right)}=16G_{\theta x}^{\left(m\right)}l_{3}^{2},$ and $G_{ij}^{\left(m\right)}$ represents the shear modulus related to the *i-j* surface for the *m*th-microlayer The variable $l_{i}^{\left(m\right)}$ is the material length scale parameter related to the *kj*-surface for the *m*th-microlayer. When the isotropic material is considered, the above-mentioned $b_{kk}^{\left(m\right)}\;(k=1-3)$ are reduced as $b_{11}^{\left(m\right)}=b_{22}^{\left(m\right)}=b_{33}^{\left(m\right)}$$=16Gl^{2},$ for which *l* denotes the material length scale parameter.

It should be noticed that the value of the material length scale parameter (*l*) in the CCST is one-half of the value of the material length scale parameter ($\hat{l}$) in the MCST. The above relation is because the relationship between the couple stress tensor and the skew-symmetric part of the curvature tensor is $\mu =-8Gl^{2}\kappa$ in the CCST, while the relationship between the couple stress tensor and the symmetric part of the curvature tensor is $m=2G{\hat{l}}^{2}\chi$. For example, the value of $\hat{l}$ of the epoxy material is $\hat{l}$ = 17.6 × 10^−6^ m, and that of *l* is *l* = 8.8 × 10^−6^ m.

The strain–displacement relationships for each microlayer are given as follows:(23)$$\varepsilon_{xx}^{\left(m\right)}=u_{x,x}^{\left(m\right)}=\sum_{i=1}^{n_{d}}\psi_{i}^{\left(m\right)}d_{ui,x}^{\left(m\right)},$$
(24)$$\varepsilon_{\theta \theta }^{\left(m\right)}=\left(1/r\right)u_{\theta ,\theta }^{\left(m\right)}+\left(1/r\right)u_{r}^{\left(m\right)}=\sum_{i=1}^{n_{d}}\left(1/r\right)\psi_{i}^{\left(m\right)}d_{vi,\theta }^{\left(m\right)}+\sum_{i=1}^{n_{d}}\left(1/r\right)\psi_{i}^{\left(m\right)}d_{wi}^{\left(m\right)},$$
(25)$$\varepsilon_{rr}^{\left(m\right)}=u_{r,r}^{\left(m\right)}=\sum_{i=1}^{n_{d}}\left(D\psi_{i}^{\left(m\right)}\right)d_{wi}^{\left(m\right)},$$
(26)$$\gamma_{xr}^{\left(m\right)}=u_{x,r}^{\left(m\right)}+u_{r,x}^{\left(m\right)}=\sum_{i=1}^{n_{d}}\left(D\psi_{i}^{\left(m\right)}\right)d_{ui}^{\left(m\right)}+\sum_{i=1}^{n_{d}}\psi_{i}^{\left(m\right)}d_{wi,x}^{\left(m\right)},$$
(27)$$\begin{array}{ll} \gamma_{\theta r}^{\left(m\right)} & =u_{\theta ,r}^{\left(m\right)}-\left(1/r\right)u_{\theta }^{\left(m\right)}+\left(1/r\right)u_{r,\theta }^{\left(m\right)}\\ & =\sum_{i=1}^{n_{d}}\left(D\psi_{i}^{\left(m\right)}\right)d_{vi}^{\left(m\right)}-\left(1/r\right)\sum_{i=1}^{n_{d}}\psi_{i}^{\left(m\right)}d_{vi}^{\left(m\right)}+\left(1/r\right)\sum_{i=1}^{n_{d}}\psi_{i}^{\left(m\right)}d_{wi,\theta }^{\left(m\right)}, \end{array}$$
(28)$$\gamma_{x\theta }^{\left(m\right)}=\left(1/r\right)u_{x,\theta }^{\left(m\right)}+u_{\theta ,x}^{\left(m\right)}=\left(1/r\right)\sum_{i=1}^{n_{d}}\psi_{i}^{\left(m\right)}d_{ui,\theta }^{\left(m\right)}+\sum_{i=1}^{n_{d}}\psi_{i}^{\left(m\right)}d_{vi,x}^{\left(m\right)},$$
where *m* = 1, 2, …, *n_l_*; the commas denote partial differentiation with respect to the suffix variables; and $D\psi_{i}^{\left(m\right)}=d\left(\psi_{i}^{\left(m\right)}\right)/dr=d\left(\psi_{i}^{\left(m\right)}\right)/d\zeta =d\left(\psi_{i}^{\left(m\right)}\right)/dz_{m}.$ The skew–symmetric parts of the curvatures–to–the–displacements relations for each microlayer are given by:(29)$$\begin{array}{cc} \kappa_{x}^{\left(m\right)} & =\left[\left(\theta_{r,\theta }^{\left(m\right)}/r\right)-\left(\theta_{\theta }^{\left(m\right)}/r\right)-\theta_{\theta ,r}^{\left(m\right)}\right]/2\\ & =\left(1/4\right)[-\left(1/r^{2}\right)u_{x,\theta \theta }^{\left(m\right)}-\left(1/r\right)u_{x,r}^{\left(m\right)}-u_{x,rr}^{\left(m\right)}+\left(1/r\right)u_{\theta ,x\theta }^{\left(m\right)}+\left(1/r\right)u_{r,x}^{\left(m\right)}+u_{r,xr}^{\left(m\right)}]\\ & =\left(1/4\right)[-\left(1/r^{2}\right)\sum_{i=1}^{n_{d}}\psi_{i}^{\left(m\right)}d_{ui,\theta \theta }^{\left(m\right)}-\left(1/r\right)\sum_{i=1}^{n_{d}}\left(D\psi_{i}^{\left(m\right)}\right)d_{ui}^{\left(m\right)}-\sum_{i=1}^{n_{d}}\left(D^{2}\psi_{i}^{\left(m\right)}\right)d_{ui}^{\left(m\right)}\\ & \;\;\;\;\;+\left(1/r\right)\sum_{i=1}^{n_{d}}\psi_{i}^{\left(m\right)}d_{vi,x\theta }^{\left(m\right)}+\left(1/r\right)\sum_{i=1}^{n_{d}}\psi_{i}^{\left(m\right)}d_{wi,x}^{\left(m\right)}+\sum_{i=1}^{n_{d}}\left(D\psi_{i}^{\left(m\right)}\right)d_{wi,x}^{\left(m\right)}], \end{array}$$
(30)$$\begin{array}{cc} \kappa_{\theta }^{\left(m\right)} & =\left(\theta_{x,r}^{\left(m\right)}-\theta_{r,x}^{\left(m\right)}\right)/2\\ & =\left(1/4\right)[\left(1/r\right)u_{x,x\theta }^{\left(m\right)}-u_{\theta ,xx}^{\left(m\right)}-\left(1/r\right)u_{\theta ,r}^{\left(m\right)}-u_{\theta ,rr}^{\left(m\right)}+\left(1/r^{2}\right)u_{\theta }^{\left(m\right)}-\left(1/r^{2}\right)u_{r,\theta }^{\left(m\right)}+\left(1/r\right)u_{r,\theta r}^{\left(m\right)}]\\ & =\left(1/4\right)[\left(1/r\right)\sum_{i=1}^{n_{d}}\psi_{i}^{\left(m\right)}d_{ui,x\theta }^{\left(m\right)}-\sum_{i=1}^{n_{d}}\psi_{i}^{\left(m\right)}d_{vi,xx}^{\left(m\right)}-\left(1/r\right)\sum_{i=1}^{n_{d}}\left(D\psi_{i}^{\left(m\right)}\right)d_{vi}^{\left(m\right)}-\sum_{i=1}^{n_{d}}\left(D^{2}\psi_{i}^{\left(m\right)}\right)d_{vi}^{\left(m\right)}\\ & \;\;\;\;\;+\left(1/r^{2}\right)\sum_{i=1}^{n_{d}}\psi_{i}^{\left(m\right)}d_{vi}^{\left(m\right)}-\left(1/r^{2}\right)\sum_{i=1}^{n_{d}}\psi_{i}^{\left(m\right)}d_{wi,\theta }^{\left(m\right)}+\left(1/r\right)\sum_{i=1}^{n_{d}}\left(D\psi_{i}^{\left(m\right)}\right)d_{wi,\theta }^{\left(m\right)}], \end{array}$$
(31)$$\begin{array}{cc} \kappa_{r}^{\left(m\right)} & =\left[\theta_{\theta ,x}^{\left(m\right)}-\left(1/r\right)\theta_{x,\theta }^{\left(m\right)}\right]/2\\ & =\left(1/4\right)[u_{x,xr}^{\left(m\right)}+\left(1/r^{2}\right)u_{\theta ,\theta }^{\left(m\right)}+\left(1/r\right)u_{\theta ,\theta r}^{\left(m\right)}-u_{r,xx}^{\left(m\right)}-\left(1/r^{2}\right)u_{r,\theta \theta }^{\left(m\right)}]\\ & =\left(1/4\right)[\sum_{i=1}^{n_{d}}\left(D\psi_{i}^{\left(m\right)}\right)d_{ui,x}^{\left(m\right)}+\left(1/r^{2}\right)\sum_{i=1}^{n_{d}}\psi_{i}^{\left(m\right)}d_{vi,\theta }^{\left(m\right)}+\left(1/r\right)\sum_{i=1}^{n_{d}}\left(D\psi_{i}^{\left(m\right)}\right)d_{vi,\theta }^{\left(m\right)}\\ & \;\;\;\;\;-\sum_{i=1}^{n_{d}}\psi_{i}^{\left(m\right)}d_{wi,xx}^{\left(m\right)}-\left(1/r^{2}\right)\sum_{i=1}^{n_{d}}\psi_{i}^{\left(m\right)}d_{wi,\theta \theta }^{\left(m\right)}], \end{array}$$
where $D^{2}\psi_{i}^{\left(m\right)}=d^{2}\psi_{i}^{\left(m\right)}/dr^{2}=d^{2}\psi_{i}^{\left(m\right)}/d\zeta^{2}=d^{2}\psi_{i}^{\left(m\right)}/dz_{m}^{2}.$

### 4.2. Hamilton’s Principle

The Euler-Lagrange equations to address the issue that we are discussing now are derived using Hamilton’s principle, and its corresponding energy functional is expressed as follows:(32)$$I=\sum_{m=1}^{n_{l}}\int_{t_{1}}^{t_{2}}\left(T^{\left(m\right)}-U_{s}^{\left(m\right)}\right)dt$$
where $T^{\left(m\right)}\;\mathrm{and}\;U_{s}^{\left(m\right)}$ denote the kinetic and strain energy of a typical *m*th-microlayer, respectively, and they are given as follows:(33)$$T^{\left(m\right)}=\int_{\zeta_{m-1}}^{\zeta_{m}}\iint_{\Omega }\left(\rho^{\left(m\right)}/2\right)\left[{\left(u_{x,t}^{\left(m\right)}\right)}^{2}+{\left(u_{\theta ,t}^{\left(m\right)}\right)}^{2}+{\left(u_{r,t}^{\left(m\right)}\right)}^{2}\right]r\;dx\;d\theta \;d\zeta ,$$
(34)$$\begin{array}{l} U_{s}^{\left(m\right)}=\left(1/2\right)\int_{\zeta_{m-1}}^{\zeta_{m}}\iint_{\Omega }[\sigma_{\left(xx\right)}^{\left(m\right)}\varepsilon_{xx}^{\left(m\right)}+\sigma_{\left(\theta \theta \right)}^{\left(m\right)}\varepsilon_{\theta \theta }^{\left(m\right)}+\sigma_{\left(rr\right)}^{\left(m\right)}\varepsilon_{rr}^{\left(m\right)}+\sigma_{\left(xr\right)}^{\left(m\right)}\gamma_{xr}^{\left(m\right)}+\sigma_{\left(\theta r\right)}^{\left(m\right)}\gamma_{\theta r}^{\left(m\right)}+\sigma_{\left(x\theta \right)}^{\left(m\right)}\gamma_{x\theta }^{\left(m\right)}\\ \;\;\;\;\;\;\;\;\;\;-2\mu_{x}^{\left(m\right)}\kappa_{x}^{\left(m\right)}-2\mu_{\theta }^{\left(m\right)}\kappa_{\theta }^{\left(m\right)}-2\mu_{r}^{\left(m\right)}\kappa_{r}^{\left(m\right)}]r\;dx\;d\theta \;d\zeta , \end{array}$$
where Ω denotes the domain of the cylindrical microshell on the x−θ surface.

As mentioned above, we take the elastic displacement components as the primary variables subject to variation. Therefore, we first perform the first-order variation of the kinetic and strain energy based on the generalized kinematics assumptions given in Equations (3)–(5), and then employ the integration by parts technique, which results in the following equations:(35)$$\delta U_{s}^{\left(m\right)}=\iint_{\Omega }\int_{\zeta_{m-1}}^{\zeta_{m}}\left\{{\left(\delta \varepsilon_{n}^{\left(m\right)}\right)}^{T}\sigma_{n}^{\left(m\right)}+{\left(\delta \varepsilon_{s}^{\left(m\right)}\right)}^{T}\sigma_{s}^{\left(m\right)}-2{\left(\delta \kappa^{\left(m\right)}\right)}^{T}\mu^{\left(m\right)}\right\}r\;dx\;d\theta \;d\zeta ,$$
(36)$$\delta T^{\left(m\right)}=-\iint_{\Omega }\int_{\zeta_{m-1}}^{\zeta_{m}}\rho^{\left(m\right)}\left[{\left(\delta u^{\left(m\right)}\right)}^{T}{\left(B_{8}^{\left(m\right)}\right)}^{T}B_{8}^{\left(m\right)}u^{\left(m\right)}{,}_{tt}+{\left(\delta w^{\left(m\right)}\right)}^{T}{\left(B_{9}^{\left(m\right)}\right)}^{T}B_{9}^{\left(m\right)}w^{\left(m\right)}{,}_{tt}\right]r\;dx\;d\theta \;d\zeta ,$$
where the superscript *T* denotes the transposition of the matrices or vectors, and
$$\varepsilon_{n}^{\left(m\right)}={\left[\begin{array}{ccc} \varepsilon_{xx}^{\left(m\right)} & \varepsilon_{\theta \theta }^{\left(m\right)} & \varepsilon_{rr}^{\left(m\right)} \end{array}\right]}^{T}=B_{1}^{\left(m\right)}u^{\left(m\right)}+B_{2}^{\left(m\right)}w^{\left(m\right)},\;\varepsilon_{s}^{\left(m\right)}={\left[\begin{array}{cc} \gamma_{\theta r}^{\left(m\right)} & \gamma_{xr}^{\left(m\right)}\;\gamma_{x\theta }^{\left(m\right)} \end{array}\right]}^{T}=B_{3}^{\left(m\right)}u^{\left(m\right)}+B_{4}^{\left(m\right)}w^{\left(m\right)},\;\kappa^{\left(m\right)}={\left[\begin{array}{cc} \kappa_{x}^{\left(m\right)} & \kappa_{\theta }^{\left(m\right)}\;\kappa_{r}^{\left(m\right)} \end{array}\right]}^{T}=B_{5}^{\left(m\right)}u^{\left(m\right)}+B_{6}^{\left(m\right)}w^{\left(m\right)},\;u^{\left(m\right)}=\left[\begin{array}{c} d_{ui}^{\left(m\right)}\\ d_{vi}^{\left(m\right)} \end{array}\right],\;w^{\left(m\right)}=\left[d_{wi}^{\left(m\right)}\right],\;\sigma_{n}^{\left(m\right)}={\left[\begin{array}{ccc} \sigma_{\left(xx\right)}^{\left(m\right)} & \sigma_{\left(\theta \theta \right)}^{\left(m\right)} & \sigma_{\left(rr\right)}^{\left(m\right)} \end{array}\right]}^{T}=\left(Q_{cn}^{\left(m\right)}B_{1}^{\left(m\right)}+Q_{dn}^{\left(m\right)}B_{5}^{\left(m\right)}\right)u^{\left(m\right)}+\left(Q_{cn}^{\left(m\right)}B_{2}^{\left(m\right)}+Q_{dn}^{\left(m\right)}B_{6}^{\left(m\right)}\right)w^{\left(m\right)},\;\sigma_{s}^{\left(m\right)}={\left[\begin{array}{cc} \sigma_{\theta r}^{\left(m\right)} & \sigma_{xr}^{\left(m\right)}\;\sigma_{x\theta }^{\left(m\right)} \end{array}\right]}^{T}=\left(Q_{cs}^{\left(m\right)}B_{3}^{\left(m\right)}+Q_{ds}^{\left(m\right)}B_{5}^{\left(m\right)}\right)u^{\left(m\right)}+\left(Q_{cs}^{\left(m\right)}B_{4}^{\left(m\right)}+Q_{ds}^{\left(m\right)}B_{6}^{\left(m\right)}\right)w^{\left(m\right)},\;\begin{array}{l} \mu^{\left(m\right)}={\left[\begin{array}{ccc} \mu_{x}^{\left(m\right)} & \mu_{\theta }^{\left(m\right)} & \mu_{r}^{\left(m\right)} \end{array}\right]}^{T}=-\left(1/2\right)\{\left[{\left(Q_{dn}^{\left(m\right)}\right)}^{T}B_{1}^{\left(m\right)}+{\left(Q_{ds}^{\left(m\right)}\right)}^{T}B_{3}^{\left(m\right)}+Q_{b}^{\left(m\right)}B_{5}^{\left(m\right)}\right]u^{\left(m\right)}\\ \;\;\;\;\;\;\;\;\;\;\;\;\;\;\;\;\;+\left[{\left(Q_{dn}^{\left(m\right)}\right)}^{T}B_{2}^{\left(m\right)}+{\left(Q_{ds}^{\left(m\right)}\right)}^{T}B_{4}^{\left(m\right)}+Q_{b}^{\left(m\right)}B_{6}^{\left(m\right)}\right]w^{\left(m\right)}\}, \end{array}\;Q_{cn}^{\left(m\right)}=\left[\begin{array}{ccc} c_{11}^{\left(m\right)} & c_{12}^{\left(m\right)} & c_{13}^{\left(m\right)}\\ c_{12}^{\left(m\right)} & c_{22}^{\left(m\right)} & c_{23}^{\left(m\right)}\\ c_{13}^{\left(m\right)} & c_{23}^{\left(m\right)} & c_{33}^{\left(m\right)} \end{array}\right],\;Q_{cs}^{\left(m\right)}=\left[\begin{array}{ccc} c_{44}^{\left(m\right)} & 0 & 0\\ 0 & c_{55}^{\left(m\right)} & 0\\ 0 & 0 & c_{66}^{\left(m\right)} \end{array}\right],\;Q_{b}^{\left(m\right)}=\left[\begin{array}{ccc} b_{11}^{\left(m\right)} & 0 & 0\\ 0 & b_{22}^{\left(m\right)} & 0\\ 0 & 0 & b_{33}^{\left(m\right)} \end{array}\right],\;B_{1}^{\left(m\right)}=\left[\begin{array}{cc} \psi_{i}^{\left(m\right)}\partial_{x} & 0\\ 0 & \left(\psi_{i}^{\left(m\right)}/r\right)\partial_{\theta }\\ 0 & 0 \end{array}\right],\;\;\;\;\;\;\;\;\;\;B_{2}^{\left(m\right)}=\left[\begin{array}{c} 0\\ \psi_{i}^{\left(m\right)}/r\\ D\psi_{i}^{\left(m\right)} \end{array}\right],\;B_{3}^{\left(m\right)}=\left[\begin{array}{cc} 0 & D\psi_{i}^{\left(m\right)}-\left(\psi_{i}^{\left(m\right)}/r\right)\\ D\psi_{i}^{\left(m\right)} & 0\\ \left(\psi_{i}^{\left(m\right)}/r\right)\partial_{\theta } & \psi_{i}^{\left(m\right)}\partial_{x} \end{array}\right],\;\;\;\;\;\;\;\;\;\;B_{4}^{\left(m\right)}=\left[\begin{array}{c} \left(\psi_{i}^{\left(m\right)}/r\right)\partial_{\theta }\\ \psi_{i}^{\left(m\right)}\partial_{x}\\ 0 \end{array}\right],\;B_{5}^{\left(m\right)}=\left(1/4\right)\left[\begin{array}{cc} -\left[\left(\psi_{i}^{\left(m\right)}/r^{2}\right)\partial_{\theta \theta }+\left(1/r\right)D\psi_{i}^{\left(m\right)}+D^{2}\psi_{i}^{\left(m\right)}\right] & \left(\psi_{i}^{\left(m\right)}/r\right)\partial_{x\theta }\\ \left(\psi_{i}^{\left(m\right)}/r\right)\partial_{x\theta } & -\left[\psi_{i}^{\left(m\right)}\partial_{xx}+\left(1/r\right)D\psi_{i}^{\left(m\right)}+D^{2}\psi_{i}^{\left(m\right)}-\left(\psi_{i}^{\left(m\right)}/r^{2}\right)\right]\\ \left(D\psi_{i}^{\left(m\right)}\right)\partial_{x} & \left(\psi_{i}^{\left(m\right)}/r^{2}\right)\partial_{\theta }+\left(D\psi_{i}^{\left(m\right)}/r\right)\partial_{\theta } \end{array}\right],\;B_{6}^{\left(m\right)}=\left(1/4\right)\left[\begin{array}{c} \left(\psi_{i}^{\left(m\right)}/r\right)\partial_{x}+\left(D\psi_{i}^{\left(m\right)}\right)\partial_{x}\\ -\left(\psi_{i}^{\left(m\right)}/r^{2}\right)\partial_{\theta }+\left(D\psi_{i}^{\left(m\right)}/r\right)\partial_{\theta }\\ -\left[\psi_{i}^{\left(m\right)}\partial_{xx}+\left(\psi_{i}^{\left(m\right)}/r^{2}\right)\partial_{\theta \theta }\right] \end{array}\right],\;B_{7}^{\left(m\right)}=\left[\begin{array}{cc} \psi_{i}^{\left(m\right)} & 0\\ 0 & \psi_{i}^{\left(m\right)} \end{array}\right],\;B_{8}^{\left(m\right)}=\left[\psi_{i}^{\left(m\right)}\right],$$
where $i=1,2,\cdots ,n_{d}$, and $m=1,2,\cdots ,n_{l}$.

### 4.3. Hermitian C^2^ Finite Element Equations

In this section, we develop weak form-based Hermitian *C*^2^ FEMs for analyzing the free vibration characteristics of simply-supported FG-GPLRC cylindrical microshells.

The boundary conditions of each microlayer at two edges of the cylindrical microshell are taken to be completely simple supports with traction loads prescribed to be zero, and they are specified as follows:(37)$$u_{\theta }^{\left(m\right)}=u_{r}^{\left(m\right)}=\sigma_{xx}^{\left(m\right)}=0\;\mathrm{at}\;x=0,\;x=L,\;\mathrm{and}\;m=1,2,\ldots ,n_{l}.$$

The primary variables of each microlayer given in Equations (17)–(19) are further expanded as a double Fourier series in the spatial domain and as harmonic functions in the time domain such that the boundary conditions of the simply-supported edges are exactly satisfied, and they are expressed as follows:(38)$$\left[u_{i}^{\left(m\right)}\theta_{ui}^{\left(m\right)}\kappa_{ui}^{\left(m\right)}\right]=\sum_{\hat{m}=1}^{\infty }\sum_{\hat{n}=0}^{\infty }\left[{\left(u_{\hat{m}\hat{n}}^{\left(m\right)}\right)}_{i}{\left(\theta_{u\hat{m}\hat{n}}^{\left(m\right)}\right)}_{i}{\left(\kappa_{u\hat{m}\hat{n}}^{\left(m\right)}\right)}_{i}\right]\cos\tilde{m}x\cos\hat{n}\theta e^{i\omega t},$$
(39)$$\left[v_{i}^{\left(m\right)}\theta_{vi}^{\left(m\right)}\kappa_{vi}^{\left(m\right)}\right]=\sum_{\hat{m}=1}^{\infty }\sum_{\hat{n}=0}^{\infty }\left[{\left(v_{\hat{m}\hat{n}}^{\left(m\right)}\right)}_{i}{\left(\theta_{v\hat{m}\hat{n}}^{\left(m\right)}\right)}_{i}{\left(\kappa_{v\hat{m}\hat{n}}^{\left(m\right)}\right)}_{i}\right]\sin\tilde{m}x\sin\hat{n}\theta e^{i\omega t},$$
(40)$$\left[w_{i}^{\left(m\right)}\theta_{wi}^{\left(m\right)}\kappa_{wi}^{\left(m\right)}\right]=\sum_{\hat{m}=1}^{\infty }\sum_{\hat{n}=0}^{\infty }\left[{\left(w_{\hat{m}\hat{n}}^{\left(m\right)}\right)}_{i}{\left(\theta_{w\hat{m}\hat{n}}^{\left(m\right)}\right)}_{i}{\left(\kappa_{w\hat{m}\hat{n}}^{\left(m\right)}\right)}_{i}\right]\sin\tilde{m}x\cos\hat{n}\theta e^{i\omega t},$$
where ω denotes the natural frequency of the FG-GPLRC cylindrical microshell; $\tilde{m}=\hat{m}\pi /L_{x}$, and $\hat{m}$ and $\hat{n}$ are the half-wave and full-wave numbers in *x* and θ directions, respectively, and $\hat{m}$ is positive integers and $\hat{n}$ is either zero or positive integers.

The element equations for the free vibration problems of the FG-GPLRC cylindrical microshell can be obtained by introducing Equations (33)–(35) into Equation (32) and then using Hamilton’s principle (i.e., δI=0), which leads to the following equation:(41)$$\sum_{m=1}^{n_{l}}\left\{\left[\begin{array}{cc} K_{II}^{\left(m\right)} & K_{I\mathrm{II}}^{\left(m\right)}\\ K_{\mathrm{II}I}^{\left(m\right)} & K_{\mathrm{II}\mathrm{II}}^{\left(m\right)} \end{array}\right]-\omega^{2}\left[\begin{array}{cc} M_{II}^{\left(m\right)} & 0\\ 0 & M_{\mathrm{II}\mathrm{II}}^{\left(m\right)} \end{array}\right]\right\}\left[\begin{array}{c} {\tilde{u}}^{\left(m\right)}\\ {\tilde{w}}^{\left(m\right)} \end{array}\right]=\left[\begin{array}{c} 0\\ 0 \end{array}\right],$$
where $K_{kl}^{\left(m\right)}\left(\psi_{i}^{\left(m\right)},\psi_{j}^{\left(m\right)}\right)={\left[K_{lk}^{\left(m\right)}\left(\psi_{j}^{\left(m\right)},\psi_{i}^{\left(m\right)}\right)\right]}^{T}$ (k,l=I and II),
$$K_{II}^{\left(m\right)}=\int_{\zeta_{m-1}}^{\zeta_{m}}\left\{{\left({\tilde{B}}_{1}^{\left(m\right)}\right)}^{T}Q_{cn}^{\left(m\right)}{\tilde{B}}_{1}^{\left(m\right)}+{\left({\tilde{B}}_{3}^{\left(m\right)}\right)}^{T}Q_{cs}^{\left(m\right)}{\tilde{B}}_{3}^{\left(m\right)}+{\left({\tilde{B}}_{5}^{\left(m\right)}\right)}^{T}Q_{b}^{\left(m\right)}{\tilde{B}}_{5}^{\left(m\right)}\right\}d\zeta ,\;K_{I\mathrm{II}}^{\left(m\right)}=\int_{\zeta_{m-1}}^{\zeta_{m}}\left\{{\left({\tilde{B}}_{1}^{\left(m\right)}\right)}^{T}Q_{cn}^{\left(m\right)}B_{2}^{\left(m\right)}+{\left({\tilde{B}}_{3}^{\left(m\right)}\right)}^{T}Q_{cs}^{\left(m\right)}{\tilde{B}}_{4}^{\left(m\right)}+{\left({\tilde{B}}_{5}^{\left(m\right)}\right)}^{T}Q_{b}^{\left(m\right)}{\tilde{B}}_{6}^{\left(m\right)}\right\}d\zeta ,\;K_{\mathrm{II}\mathrm{II}}^{\left(m\right)}=\int_{\zeta_{m-1}}^{\zeta_{m}}\left\{{\left(B_{2}^{\left(m\right)}\right)}^{T}Q_{cn}^{\left(m\right)}B_{2}^{\left(m\right)}+{\left({\tilde{B}}_{4}^{\left(m\right)}\right)}^{T}Q_{cs}^{\left(m\right)}{\tilde{B}}_{4}^{\left(m\right)}+{\left({\tilde{B}}_{6}^{\left(m\right)}\right)}^{T}Q_{b}^{\left(m\right)}{\tilde{B}}_{6}^{\left(m\right)}\right\}d\zeta ,\;M_{II}^{\left(m\right)}=\int_{\zeta_{m-1}}^{\zeta_{m}}{\left(B_{7}^{\left(m\right)}\right)}^{T}\rho^{\left(m\right)}B_{7}^{\left(m\right)}d\zeta ,\;M_{\mathrm{II}\mathrm{II}}^{\left(m\right)}=\int_{\zeta_{m-1}}^{\zeta_{m}}{\left(B_{8}^{\left(m\right)}\right)}^{T}\rho^{\left(m\right)}B_{8}^{\left(m\right)}d\zeta ;\;{\tilde{B}}_{1}^{\left(m\right)}=\left[\begin{array}{cc} -\tilde{m}\psi_{i}^{\left(m\right)} & 0\\ 0 & \hat{n}\psi_{i}^{\left(m\right)}/r\\ 0 & 0 \end{array}\right],{\tilde{B}}_{3}^{\left(m\right)}=\left[\begin{array}{cc} 0 & D\psi_{i}^{\left(m\right)}-\left(\psi_{i}^{\left(m\right)}/r\right)\\ D\psi_{i}^{\left(m\right)} & 0\\ -\hat{n}\psi_{i}^{\left(m\right)}/r & \tilde{m}\psi_{i}^{\left(m\right)} \end{array}\right],{\tilde{B}}_{4}^{\left(m\right)}=\left[\begin{array}{c} -\hat{n}\psi_{i}^{\left(m\right)}/r\\ \tilde{m}\psi_{i}^{\left(m\right)}\\ 0 \end{array}\right],\;{\tilde{B}}_{5}^{\left(m\right)}=\left(1/4\right)\left[\begin{array}{cc} \left[{\hat{n}}^{2}\psi_{i}^{\left(m\right)}/r^{2}-\left(1/r\right)D\psi_{i}^{\left(m\right)}-D^{2}\psi_{i}^{\left(m\right)}\right] & \tilde{m}\hat{n}\psi_{i}^{\left(m\right)}/r\\ \tilde{m}\hat{n}\psi_{i}^{\left(m\right)}/r & \left[{\tilde{m}}^{2}\psi_{i}^{\left(m\right)}-\left(1/r\right)D\psi_{i}^{\left(m\right)}-D^{2}\psi_{i}^{\left(m\right)}+\left(\psi_{i}^{\left(m\right)}/r^{2}\right)\right]\\ -\tilde{m}D\psi_{i}^{\left(m\right)} & \left(\hat{n}\psi_{i}^{\left(m\right)}/r^{2}\right)+\left(\hat{n}D\psi_{i}^{\left(m\right)}/r\right) \end{array}\right],\;{\tilde{B}}_{6}^{\left(m\right)}=\left(1/4\right)\left[\begin{array}{c} \tilde{m}\left(\psi_{i}^{\left(m\right)}/r\right)+\tilde{m}\left(D\psi_{i}^{\left(m\right)}\right)\\ \left(\hat{n}\psi_{i}^{\left(m\right)}/r^{2}\right)-\hat{n}\left(D\psi_{i}^{\left(m\right)}/r\right)\\ \left[{\tilde{m}}^{2}\psi_{i}^{\left(m\right)}+{\hat{n}}^{2}\psi_{i}^{\left(m\right)}/r^{2}\right] \end{array}\right],{\tilde{u}}^{\left(m\right)}=\left[\begin{array}{c} {\left(d_{u\hat{m}\hat{n}}^{\left(m\right)}\right)}_{i}\\ {\left(d_{v\hat{m}\hat{n}}^{\left(m\right)}\right)}_{i} \end{array}\right]=\left[\begin{array}{c} \begin{array}{l} {\left(u_{\hat{m}\hat{n}}^{\left(m\right)}\right)}_{i}\\ {\left(\theta_{u\hat{m}\hat{n}}^{\left(m\right)}\right)}_{i}\\ {\left(\kappa_{u\hat{m}\hat{n}}^{\left(m\right)}\right)}_{i} \end{array}\\ \begin{array}{l} {\left(v_{\hat{m}\hat{n}}^{\left(m\right)}\right)}_{i}\\ {\left(\theta_{v\hat{m}\hat{n}}^{\left(m\right)}\right)}_{i}\\ {\left(\kappa_{v\hat{m}\hat{n}}^{\left(m\right)}\right)}_{i} \end{array} \end{array}\right],\;{\tilde{w}}^{\left(m\right)}=\left[{\left(d_{w\hat{m}\hat{n}}^{\left(m\right)}\right)}_{i}\right]=\left[\begin{array}{l} {\left(w_{\hat{m}\hat{n}}^{\left(m\right)}\right)}_{i}\\ {\left(\theta_{w\hat{m}\hat{n}}^{\left(m\right)}\right)}_{i}\\ {\left(\kappa_{w\hat{m}\hat{n}}^{\left(m\right)}\right)}_{i} \end{array}\right].$$

Assembling the element stiffness matrix and the element mass matrix for each microlayer leads to the structure stiffness matrix and the structure mass matrix as follows:(42)$$\left\{\left[\begin{array}{cc} K_{11} & K_{12}\\ K_{21} & K_{22} \end{array}\right]-\omega^{2}\left[\begin{array}{cc} M_{11} & 0\\ 0 & M_{22} \end{array}\right]\right\}\left[\begin{array}{c} \tilde{u}\\ \tilde{w} \end{array}\right]=\left[\begin{array}{c} 0\\ 0 \end{array}\right].$$

Equation (42) represents the system equations for analyzing the free vibration behavior of a simply-supported FG-GPLRC cylindrical microshell. A nontrivial solution of Equation (42) exists if and only if the determinant of its coefficient matrix vanishes. Finally, the natural frequencies of the FG-GPLRC cylindrical microshell for fixed values of the wave number pair $\left(\hat{m},\hat{n}\right)$ can be obtained by solving the following equation as follows:(43)$$\left|\left[\begin{array}{cc} K_{11} & K_{12}\\ K_{21} & K_{22} \end{array}\right]-\omega^{2}\left[\begin{array}{cc} M_{11} & 0\\ 0 & M_{22} \end{array}\right]\right|=0.$$

## 5. Numerical Examples

Based on the CCST, the two-node and three-node Hermitian *C*^2^ FEMs are developed as discussed above. Then, they are applied to investigate the free vibration characteristics of simply-supported FG-GPLRC cylindrical microshells. In addition, an *h*-convergence technique is adopted for this work to achieve the convergent solutions of each Hermitian *C*^2^ FEM.

### 5.1. Homogeneous Isotropic SWCNTs

No benchmark solutions for the free vibration problem of simply-supported FG cylindrical microshells are reported in the literature. Thus, we compare the solutions of natural frequencies obtained using the Hermitian *C*^2^ FEMs with the accurate solutions of the natural frequencies for homogeneous isotropic SWCNTs reported in the literature, following Zeighampour and Beni [47], by setting the material property gradient index at $\kappa_{p}=0$ and the material length scale parameter at $\hat{l}/h$ = 0 and 1.

Table 1 shows the convergence and validation studies of the two-node Lagrangian *C*^0^ [44], Hermitian *C*^1^ [45], and Hermitian *C*^2^ FEMs for the natural frequencies of simply-supported SWCNTs. The material properties of the SWCNT are *E =* 1.06 Tpa, υ = 0.3, and $\rho \;=\;2300\;\mathrm{kg}/m^{3}$. The geometric parameters of the SWCNT are *R* = 2.32 nm, *L*/*R* = 5, and *h*/*R* = 0.05. The wave number pairs are $\left(\hat{m},\hat{n}\right)$ = (1, 1), (2, 2), (3, 3), (4, 4), and (5, 5). In addition, a dimensionless frequency parameter is defined as $\bar{\omega }=\omega R\sqrt{\rho /E}$.

It can be seen in Table 1 that the convergence rate for various FEMs is arranged in descending order: Hermitian *C*^2^ FEM > Hermitian *C*^1^ FEM > Lagrangian *C*^0^ FEM. The convergent solutions obtained using various FEMs are identical to one another. In the cases of macroshells ($\hat{l}/h$ = 0), the relative errors between the solutions obtained using the current Hermitian *C*^2^ FEM and Zeighampour and Beni’s solutions obtained using the MCST-based LCST are 0.2%, 1.6%, 3.0%, 3.0%, and 6.9% for the wave number pairs $\left(\hat{m},\hat{n}\right)$ = (1, 1), (2, 2), (3, 3), (4, 4), and (5, 5), respectively. These deviations between them are mainly due to shear deformation effects and thickness stretching effects. The relative errors for high-frequency vibration modes are greater than those for low-frequency vibration modes, which indicates that these effects on the natural frequency are more significant for high-frequency vibration modes than for low-frequency vibration modes. In the cases of microshells ($\hat{l}/h$ = 1), the relative errors increase to 7.7%, 10.5%, 5.0%, 4.1%, and 10.0% for the wave number pairs $\left(\hat{m},\hat{n}\right)$ = (1, 1), (2, 2), (3, 3), (4, 4), and (5, 5), respectively. These deviations are mainly due to 3D couple-stress tensor effects, in addition to shear deformation effects and thickness stretching effects.

### 5.2. FG Cylindrical Macroshells

In this section, we compare the solutions of natural frequencies obtained using the two-node and three-node Hermitian *C*^2^ FEMs with the exact and approximate 3D solutions of the natural frequencies for FG cylindrical macroshells reported in the literature by assigning the material length scale parameter a value of zero.

Table 2 shows the convergence and validation studies of the two-node and three-node Hermitian *C*^2^ FEMs for the natural frequencies of simply-supported FG cylindrical macroshells. The FG cylindrical macroshell of interest is considered to be composed of nickle and stainless steel. The inner surface of the macroshell is completely composed of nickel, and the outer surface of the macroshell is completely composed of stainless steel. The material properties of the macroshell are assumed to obey a power-law distribution of the volume fractions of the constituents varying in the thickness direction. The effective material properties are estimated using a rule of mixtures which are expressed as follows:(44)$$P_{eff}=P_{n}(1-\Gamma_{ss})+P_{ss}\Gamma_{ss}$$
where *P* represents some specific material properties, including Young’s modulus, Poisson’s ratio, and mass density; the subscripts *eff*, *n*, and *ss* denote the FG material, nickel, and stainless steel, respectively; $\Gamma_{ss}$ is the volume fraction function of stainless steel varying in the thickness direction, and $\Gamma_{ss}={\left[\left(r-a\right)/h\right]}^{\kappa_{p}}$, where $\kappa_{p}$ is defined as the material property gradient index; and the material properties of the constituents (i.e., nickel and stainless steel), following Loy et al. [48], are listed as:(45)$$E_{ss}=2.07788\;\times \;10^{11}\mathrm{Pa},\;\upsilon_{ss}=0.317756,\;\rho_{ss}=8166\;{\mathrm{kg}/m}^{3};$$
(46)$$E_{n}=2.05098\;\times \;10^{11}\;\mathrm{Pa},\;\upsilon_{n}=0.3100,\;\rho_{n}=8900\;{\mathrm{kg}/m}^{3}.$$

Table 2 shows the convergence and validation studies for the natural frequencies [ω/(2π)] (Hz) of a simply-supported FG cylindrical macroshell obtained using the two-node and three-node Hermitian *C*^2^ FEMs. The relevant parameters in this analysis are $\kappa_{p}$ = 0.5, 1, and 30; the wave number pairs are $\hat{m}=1$ and $\hat{n}=0$–10; and *L*/*a* = 20, and *h*/*a* = 0.002.

It can be seen in Table 2 that the convergent solutions of natural frequencies are obtained when *n_l_* = 4 is used. The results also show that the convergence rate of the three-node Hermitian *C*^2^ FEM is faster than that of the two-node Hermitian *C*^2^ FEM. The convergent solutions of natural frequencies of the FG macroshell are in excellent agreement with the solutions obtained by Liu et al. [49] with the state space method and by Loy et al. [48] with the Rayleigh-Ritz method based on the LCST. It is noted that the natural frequencies of the FG macroshell decrease when the value of $\kappa_{p}$ becomes greater. The above phenomenon observed is because an increase in the value of $\kappa_{p}$ indicates both a decrease in the volume fraction of stainless steel and an increase in the volume fraction of nickel, leading to the overall stiffness of the FG macroshell decreasing and the total mass of the FG macroshell increasing, which in turn decreases the natural frequencies of the macroshell. In addition, the fundamental frequency of the very thin FG macroshell (*R*/*h* = 500.5) occurs when the wave number pair is $\left(\hat{m},\hat{n}\right)=(1,\;3).$ Because of the excellent performance of the three-node Hermitian *C*^2^ FEM shown above, it is selected for the following free vibration analyses of laminated GPLRC cylindrical microshells and FG-GPLRC cylindrical microshells.

### 5.3. Laminated GPLRC Cylindrical Microshells

In this section, we apply the three-node Hermitian *C*^2^ FEM to investigate the free vibration characteristics of simply-supported, laminated GPLRC cylindrical macroshells (*l* = 0) and microshells (l≠0). Epoxy is used as the matrix material, and GPLs are used as the reinforcement material, which is randomly dispersed in the thickness direction. Five layer-wise constant functions of the GPL distribution patterns are considered: UD and FG V-type, A-type, X-type, and O-type distributions. The relationship between the GPL volume fraction and its weight fraction for the *m*th-layer, in the light of Song et al. [50,51], is expressed in the following forms:(47)$$V_{GPL}^{\left(m\right)}=\left(W_{GPL}^{\left(m\right)}\rho_{epoxy}\right)/\left[W_{GPL}^{\left(m\right)}\rho_{epoxy}+\rho_{GPL}\left(1-W_{GPL}^{\left(m\right)}\right)\right],$$
where the superscript *m* denotes the *m*th-layer, and *m* = 1 − $n_{l}$, with $n_{l}$ denoting the total number of the layers constituting the laminated cylindrical shell. In this work, following Liu et al. [49], the value of $n_{l}$ is assigned as $n_{l}$ = 20. Further, $W_{GPL}^{\left(m\right)}$ represents the GPL weight fraction for the *m*th-layer.

When a specific value of the total weight fraction ($W_{GPL}^{*}$) of the laminated GPLRC cylindrical shell is given, the GPL weight fractions of the *m*th-layer for different GPL distribution patterns are expressed as follows:(48)$$W_{GPL}^{\left(m\right)}=\left\{ \begin{array}{ll} W_{GPL}^{*} & (\mathrm{for}\mathrm{the}\mathrm{UD}),\\ 4W_{GPL}^{*}\left\{\left[\left(n_{l}+1\right)/2\right]-\left|m-\left[\left(n_{l}+1\right)/2\right]\right|\right\}/\left(n_{l}+2\right) & (\mathrm{for}\;\mathrm{the}\;\mathrm{FG}\;O\text{-}\mathrm{type}\;\mathrm{distribution}),\\ 4W_{GPL}^{*}\left\{\left(1/2\right)+\left|m-\left[\left(n_{l}+1\right)/2\right]\right|\right\}/\left(n_{l}+2\right) & (\mathrm{for}\;\mathrm{the}\;\mathrm{FG}\;X\text{-}\mathrm{type}\;\mathrm{distribution}),\\ W_{GPL}^{*}\left[2m/\left(n_{l}+1\right)\right] & (\mathrm{for}\;\mathrm{the}\;\mathrm{FG}\;V\text{-}\mathrm{type}\;\mathrm{distribution}),\\ W_{GPL}^{*}\left[2\left(n_{l}+1-m\right)/\left(n_{l}+1\right)\right] & (\mathrm{for}\;\mathrm{the}\;\mathrm{FG}\;A\text{-}\mathrm{type}\;\mathrm{distribution}), \end{array}\right.$$
where *m* = 1 − $n_{l}$, with *m* being counted from the bottom.

The effective Young’s modulus is estimated using the Halpin-Tsai model and is given in Equations (1)–(7), and the effective Poisson’s ratio and the effective mass density are estimated using the rule of mixtures and are given in Equations (8) and (9), respectively. Unless otherwise stated, the material properties and geometric parameters of GPL, following Liu et al. [48], are given as follows:(49)$$E_{GPL}=1.01\;\mathrm{TPa},\;\rho_{GPL}=1.06\;g/{\mathrm{cm}}^{3}=1060\;\mathrm{kg}/m^{3},\;\upsilon_{GPL}=0.186;$$
(50)$$E_{epoxy}=3.0\;\mathrm{GPa},\;\rho_{epoxy}=1.2\;g/{\mathrm{cm}}^{3}=1200\;\mathrm{kg}/m^{3},\;\upsilon_{epoxy}=0.34;$$
(51)$${\left(L_{x}\right)}_{\mathrm{GPL}}=2.5\;\mu m,\;{\left(L_{y}\right)}_{\mathrm{GPL}}=1.5\;\mu m,h_{\mathrm{GPL}}=1.5\;\mathrm{nm},\;W_{\mathrm{GPL}}^{*}=1.5\%.$$

A nondimensional natural frequency parameter $\bar{\omega }$ is defined as the same form as that used in Liu et al. [49] and is given as:(52)$$\bar{\omega }=\omega \left[R-\left(h/2\right)\right]\sqrt{E_{epoxy}/\rho_{epoxy}}.$$

Table 3 shows the solutions obtained using the three-node Hermitian *C*^2^ FEM for the first three dimensionless natural frequencies of the laminated GPLRC cylindrical macroshells $\left(\hat{l}/h=0\right)$ and microshells $\left(\hat{l}/h\neq 0\right)$ for different values of the half-wave number, $\hat{m}$, with $\hat{m}$ = 2, 5, and 10. The geometrical parameters of the laminated GPLRC cylindrical shell under observation are L/R=2π/3; R/h=3/2. It can be seen in Table 3 that in the cases of macroshells $\left(\hat{l}/h=0\right)$, the solutions of dimensionless natural frequencies obtained using the Hermitian *C*^2^ FEM are in excellent agreement with the solutions obtained by Liu et al. [49] with the state space method. The relative errors for the cases considered in Table 2 are less than 0.05%. The results also show that the dimensionless natural frequency increases when the $\hat{l}/h$ ratio increases. The above results are because an increase in the material length scale parameter causes the GPLRC microshell to become stiffer, in turn increasing its natural frequency. The results also show that in the cases of the total weight fraction $W_{GPL}^{*}=0.015$, the dimensionless natural frequency of laminated GPLRC shells is much greater than that of homogeneous epoxy shells. For a particular case of $\hat{l}/h=0.4$, and $\left(\hat{m},\hat{n}\right)=(2,1)$, the dimensionless fundamental frequencies of laminated GPLRC microshells for different GPL distribution patterns are arranged following the descending order as FG V-type > FG X-type > UD > FG O-type > FG A-type distributions. The dimensionless fundamental frequencies of the laminated GPLRC shells for various GPL distribution patterns: FG V-type, FG X-type, UD, FG O-type, and FG A-type distributions are 2.464, 2.457, 2.452, 2.362, and 2.154 times the dimensionless fundamental frequencies of homogeneous epoxy shells, respectively.

### 5.4. FG-GPLRC Cylindrical Microshells

This section applies the three-node Hermitian *C*^2^ FEM to examine the free vibration characteristics of simply-supported FG-GPLRC cylindrical microshells. Again, epoxy is used as the matrix material, and GPLs are used as the reinforcement material. Five GPL distribution patterns varying in the thickness direction are considered: UD and FG V-type, A-type, X-type, and O-type distributions. The relationship between the GPL volume fraction ($V_{GPL}\left(\zeta \right)$) and its weight fraction ($W_{GPL}\left(\zeta \right)$) for each surface in the thickness direction is given in Equation (47).

When a specific value of the total weight fraction ($W_{GPL}^{*}$) of the FG cylindrical shell is given, five GPL weight fractions varying in the thickness direction for different GPL distribution patterns are expressed in Equation (10)–(14). A nondimensional natural frequency parameter $\bar{\omega }$ is defined as follows:(53)$$\bar{\omega }=\omega h\sqrt{E_{epoxy}/\rho_{epoxy}}$$

The results in Table 4 show that the convergent solutions obtained using the three-node Hermitian *C*^2^ FEM for the lowest dimensionless natural frequencies of the FG-GPLRC microshell with different functions of GPL distribution patterns and other wave number pairs, for which *L*/*R* = 10; *R*/*h* = 5, 10, and 20; $\left(\hat{m},\hat{n}\right)$ = (1, 1), (1, 2), (2, 2), (1, 3), (2, 3), and (3, 3); and $\hat{l}/h$ = 0, 0.2, 0.4, 0.6, 0.8, and 1.0. It can be seen in Table 4 that for a moderately thick microshell (*R*/*h* = 10), the dimensionless fundamental frequencies occur at wave number pair $\left(\hat{m},\hat{n}\right)$ = (1, 1), and the effect of the material length scale parameter on the dimensionless fundamental frequencies ($\left(\hat{m},\hat{n}\right)$ = (1, 1)) is more unsignificant than that on the dimensionless natural frequencies for other wave number pairs ($\left(\hat{m},\hat{n}\right)\neq$(1, 1)). In those cases, for the V-type GPL distribution pattern, the magnitude ratios of the lowest natural frequency between the FG-GPLRC microshell ($\hat{l}/h$ = 1.0) and the FG-GPLRC macroshell ($\hat{l}/h$ = 0) are 1.0013, 2.3696, 1.8649, 2.4326, 2.4018, and 2.3036 for the wave number pairs $\left(\hat{m},\hat{n}\right)$ = (1, 1), (1, 2), (2, 2), (1, 3), (2, 3), and (3, 3), respectively. The results also show that the fundamental natural frequencies ($\left(\hat{m},\hat{n}\right)$ = (1, 1)) of various GPL distribution patterns are arranged in descending order as: FG V-type > FG X-type > UD > FG O-type > FG A-type distributions. This is because the more the GPL is dispersed away from the center of the GPLRC microshell, the stiffer the GPLRC microshell is, which in turn increases its dimensionless natural frequency.

In Figure 3, Figure 4, Figure 5, Figure 6, Figure 7, Figure 8 and Figure 9, we present a parametric study related to some key effects on the lowest dimensionless natural frequencies of simply-supported FG-GPLRC cylindrical microshells, including the effects of the ratios of $\hat{l}/h$, *R*/*h*, and *L*/*R*, the GPL weight fraction, and the length–to–thickness ratio of the GPL, and different GPL distribution patterns. The material properties of epoxy and GPLs and the geometric dimensions of GPLs are given in Equation (49)–(51), the value of the material length scale parameter is set at $\hat{l}=2l=17.6\times 10^{-6}\;m$, and the dimensionless natural frequency $\bar{\omega }$ is defined as the same form in Equation (45).

Figure 3 and Figure 4 show the variations in the lowest dimensionless natural frequencies of an FG-GPLRC thick (*R*/*h* = 5) and an FG-GPLRC thin (*R*/*h* = 20) cylindrical microshell, respectively, with different values of wave numbers for various GPL distribution patterns, for which $\hat{m}=1-3$, $\hat{n}=0-7$, *L*/*R* = 5, $W_{GPL}^{*}$ = 0.01, and $\hat{l}/h$ = 1.0. It can be seen in Figure 3 and Figure 4 that for a fixed value of $\hat{m}$, when the value of $\hat{n}$ increases from zero to seven, the dimensionless natural frequency first decreases to a relative minimum value, and then it gradually increases to a relative maximum value. The fundamental natural frequency for different GPL distribution patterns always occurs when the wave number pair is $\left(\hat{m},\hat{n}\right)$ = (1, 1) for the thick microshell (*R*/*h* = 5) and is $\left(\hat{m},\hat{n}\right)$ = (1, 2) for the thin microshell (*R*/*h* = 20).

By comparing the dimensionless natural frequency of an FG-GPLRC cylindrical microshell in Figure 3a–e and Figure 4a–e with those of a homogeneous epoxy cylindrical microshell in Figure 3f and Figure 4f, respectively, it is shown that the dimensionless natural frequency of a homogeneous epoxy cylindrical microshell increases significantly by adding a small amount of GPLs. For example: for the V-type GPL distribution pattern, by increasing the weight fraction of GPLs by 1%, the natural frequency of FG-GPLRC cylindrical microshells can be increased to 2.13 and 2.05 times that of the homogeneous epoxy cylindrical microshells for the mid-surface radius–to–thickness ratios *R*/*h* = 5 and *R*/*h* = 20, respectively.

Figure 5a,b show that the variations in the lowest dimensionless natural frequency of an FG-GPLRC cylindrical microshell with the ratio of $\hat{l}/h$ for different GPL distribution patterns. Various parameters are considered as $\hat{l}/h$ = 0–1; $\left(\hat{m},\hat{n}\right)$ = (1, 1) and (1, 2) in Figure 5a,b, respectively; *R*/*h* = 5, and *L*/*R* = 10; and $W_{GPL}^{*}$ = 0.01. It can be seen in Figure 5a that for the case of $\left(\hat{m},\hat{n}\right)$ = (1, 1), the values of the dimensionless natural frequencies for different GPL distribution patterns can be arranged in descending order as FG V-type > FG X-type > UD > FG O-type > FG A-type distributions. Likewise, they are in descending order as FG X-type > (FG A-type, UD) > (FG O-type, FG V-type) distributions for the case of $\left(\hat{m},\hat{n}\right)$ = (1, 2) in Figure 5b. The results also show that the effect of the material length scale parameter on the lowest dimensionless natural frequencies for the vibration modes with $\left(\hat{m},\hat{n}\right)$ = (1, 2), is more significant than that for those with $\left(\hat{m},\hat{n}\right)$ = (1, 1).

Figure 6 and Figure 7 show the variations in the dimensionless fundamental frequency of an FG-GPLRC cylindrical microshell with the ratios of *R*/*h* and *L*/*R*, respectively, for different GPL distribution patterns. The relevant parameters are $\left(\hat{m},\hat{n}\right)$ = (1, 1), *R*/*h* = 5–15, *L*/*R* = 10, $\hat{l}/h=1.0$, and $W_{GPL}^{*}$ = 0.01 given in Figure 6, and $\left(\hat{m},\hat{n}\right)$ = (1, 1), *R*/*h* = 5, *L*/*R* = 2–10, $\hat{l}/h=1.0$, and $W_{GPL}^{*}$ = 0.01 given in Figure 7.

It can be seen in Figure 6 and Figure 7 that the dimensionless fundamental frequency decreases when the microshell becomes thinner and longer. This observation indicated that an increase in the ratios of R/h and L/R causes the microshell to become softer, decreasing its dimensionless fundamental frequencies. The results also show that in the ranges of *R*/*h* = 5–15 and *L*/*R* = 10 considered in Figure 6 and of *R*/*h* = 5 and *L*/*R* = 2–10 considered in Figure 7, the fundamental frequency occurs when the wave number pair is $\left(\hat{m},\hat{n}\right)$ = (1, 1). Again, the values of the dimensionless natural frequencies for different GPL patterns can be arranged in descending order as FG V-type > FG X-type > UD > FG O-type > FG A-type distributions.

Figure 8a,b show the variations in the dimensionless fundamental frequency of an FG-GPLRC thick and an FG-GPLRC thin cylindrical microshell, respectively, with the weight fraction of GPL for different GPL distribution patterns.

The relevant parameters are $\hat{l}/h=1.0$, *R*/*h* = 5, *L*/*R* = 5, and $\left(\hat{m},\hat{n}\right)$ = (1, 1) in Figure 8a, and $\hat{l}/h=1.0$, *R*/*h* = 20, *L*/*R* = 5, and $\left(\hat{m},\hat{n}\right)$ = (1, 2) in Figure 8b. It can be seen in Figure 8a,b that the dimensionless fundamental frequency increases when the weight fraction of the GPL increases. This observation indicates that an increase in the weight fraction of GPL leads to an increase in the overall stiffness of the microshell and a decrease in the total mass of the microshell, increasing its dimensionless fundamental frequency.

Figure 9 shows the variations in the dimensionless fundamental frequency of an FG-GPLRC cylindrical microshell with the ratios of the length–to–thickness of GPLs, for different GPL distribution patterns. The relevant parameters are *R*/*h* = 20, *L*/*R* = 5, $\left(\hat{m},\hat{n}\right)$ = (1, 2), and $W_{GPL}^{*}$ = 0.01.

Figure 9 shows that the dimensionless fundamental frequency increases when the length–to–thickness ratio increases. This observation indicates that when the values of the length, the width, and the GPL weight fraction are fixed, an increase in the length–to–thickness ratio leads to an increase in the total contact surface area between GPL reinforcements and epoxy matrix and an increase in the overall stiffness of the microshell, which in turn increases the dimensionless fundamental frequency.

## 6. Concluding Remarks

Within a framework of the CCST, we developed a 3D weak formulation of the CCST-based Hermitian *C*^2^ FEM for analyzing the free vibration characteristics of simply-supported FG-GPLRC cylindrical microshells using Hamilton’s principle. In addition, a parametric study related to some key effects on the lowest natural frequencies of simply-supported FG-GPLRC cylindrical microshells was carried out, including the material length scale parameter, the GPL distribution patterns, the GPL weight fractions, the length–to–mid-surface radius and mid-surface radius–to–thickness ratio of the microshell, and the length–to–thickness ratio of GPLs. Some conclusions drawn from the parametric study were summarized as follows:
The CCST-based Hermitian *C*^2^ FEM was validated to be accurate and to converge rapidly by comparing the solutions it produced with the 3D exact and the quasi-3D solutions for FG cylindrical macroshell reported in the literature by assigning the length scale parameter to a value of zero.In numerical examples, the results showed that:
(a)An increase in the material length scale parameter caused the FG cylindrical microshell to become stiffer, increasing the lowest natural frequency of the microshell. The effect of the material length scale parameter on the lowest natural frequency for the high-frequency vibration modes was more significant than that for the fundamental vibration mode.(b)For moderately thick (*R*/*h* = 10) and thick (*R*/*h* = 5) cylindrical microshells, the fundamental mode of vibration always occurred when the wave number pair was $\left(\hat{m},\hat{n}\right)$ = (1, 1). In that case, the values of the dimensionless natural frequencies for different GPL distribution patterns can be arranged in descending order as follows: FG V-type > FG X-type > UD > FG O-type > FG A-type distributions.(c)For thin (*R*/*h* = 20) microshells, the fundamental mode of vibration always occurred when the wave number pair was $\left(\hat{m},\hat{n}\right)$ = (1, 2). In that case, the values of the dimensionless natural frequencies for different GPL distribution patterns can be arranged in descending order as follows: FG X-type > (FG A-type, UD) > (FG O-type, FG V-type) distributions.(d)By increasing the weight fraction of GPLs by 1%, the natural frequency of FG-GPLRC cylindrical microshells can be increased to more than twice that of the homogeneous cylindrical microshells.(e)The dimensionless fundamental frequency decreased when the microshells became thinner and longer.(f)The dimensionless fundamental frequency increased when the weight fraction of the GPL increased and when the length–to–thickness ratio of the GPL increased.


The 3D solutions for the free vibration characteristics of FG-GPLRC cylindrical microshells are relatively limited in the literature. Therefore, the current solutions can be used as a reference to evaluate the theoretical accuracy of various 2D size-dependent shear deformation theories of FG cylindrical microshells. Furthermore, due to the excellent performance of the CCST-based Hermitian *C*^2^ FEM, it is recommended to analyze various mechanical behaviors of FG elastic/piezoelectric cylindrical microshells and various FG elastic/piezoelectric microshells of revolution.

## Figures and Tables

**Figure 1 materials-16-02363-f001:**
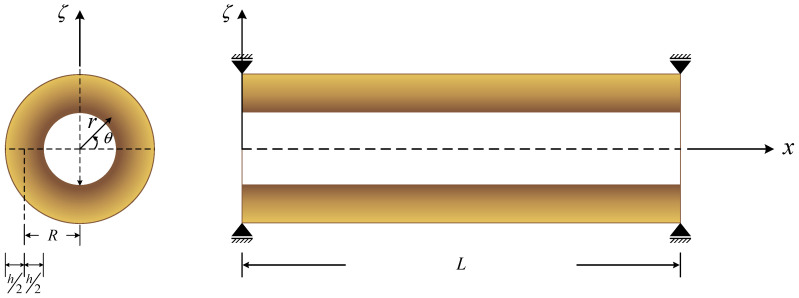
The schematic diagram of a simply-supported FG cylindrical microshell.

**Figure 2 materials-16-02363-f002:**
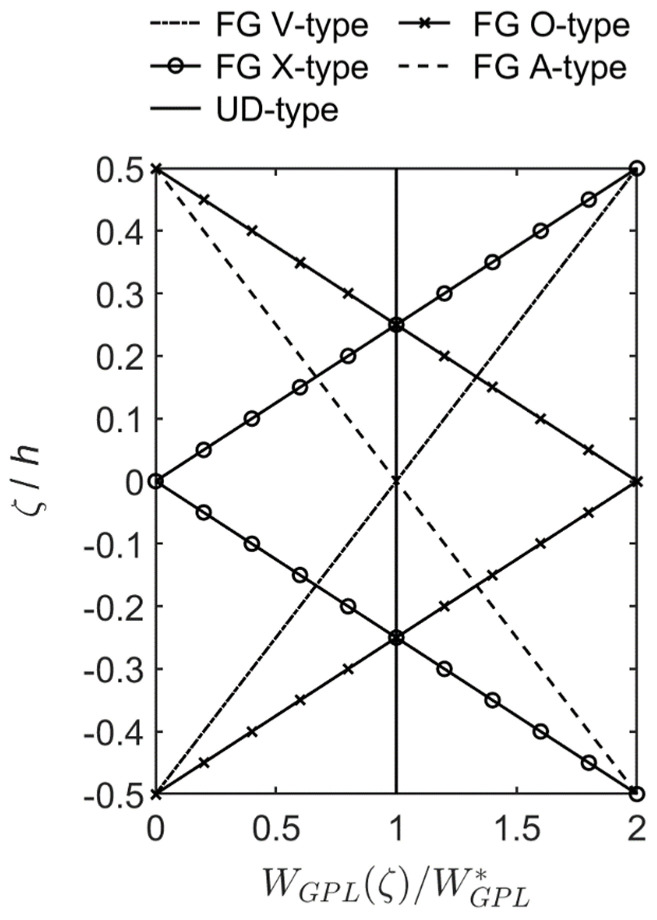
Five functions of GPL distribution patterns varying in the thickness direction of an FG-GPLRC cylindrical microshell.

**Figure 3 materials-16-02363-f003:**
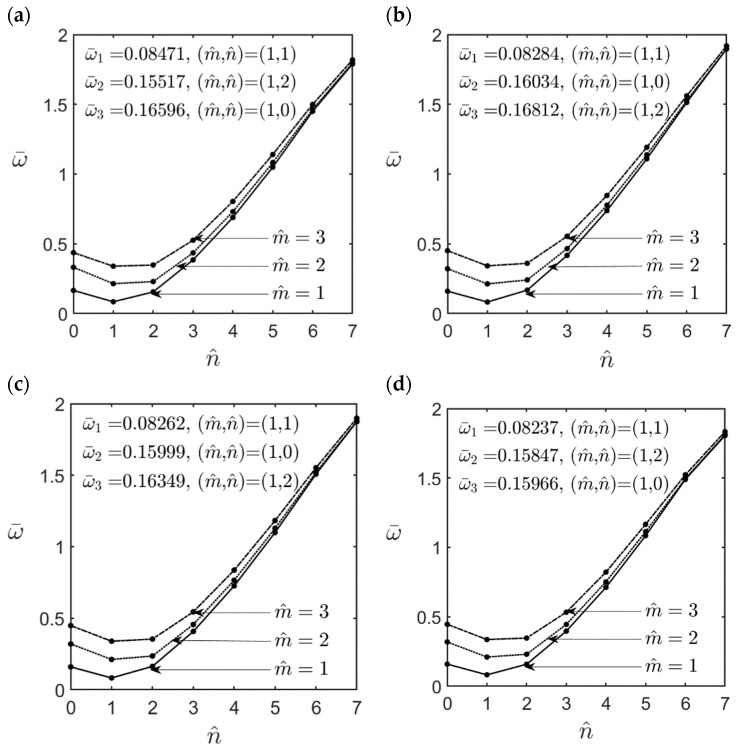
Variations in the lowest dimensionless natural frequencies of an FG-GPLRC thick cylindrical microshell with the wave number pairs $\left(\hat{m},\hat{n}\right)$, for which *R*/*h* = 5, *L*/*R* = 5, $\hat{l}/h=1.0$, $W_{GPL}^{*}=0.01$, and for the cases of (**a**) FG V-type, (**b**) FG X-type, (**c**) UD, (**d**) FG O-type, (**e**) FG A-type GPL distribution patterns, and (**f**) homogeneous epoxy.

**Figure 4 materials-16-02363-f004:**
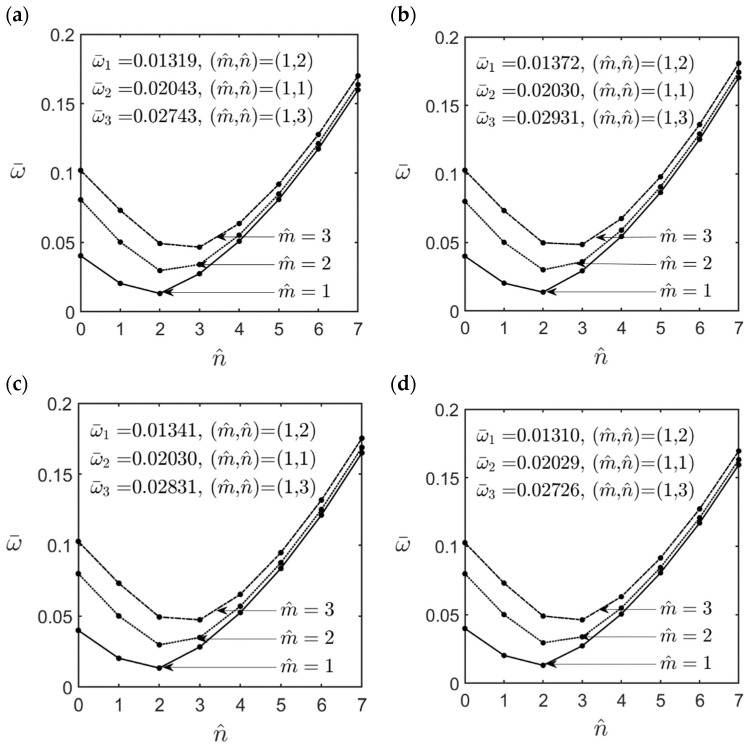
Variations in the lowest dimensionless natural frequencies of an FG-GPLRC thin cylindrical microshell with the wave number pairs $\left(\hat{m},\hat{n}\right)$, for which *R*/*h* = 20, *L*/*R* = 5, $\hat{l}/h=1.0$, $W_{GPL}^{*}=0.01$, and for the cases of (**a**) FG V-type, (**b**) FG X-type, (**c**) UD, (**d**) FG O-type, (**e**) FG A-type GPL distribution patterns, and (**f**) homogeneous epoxy.

**Figure 5 materials-16-02363-f005:**
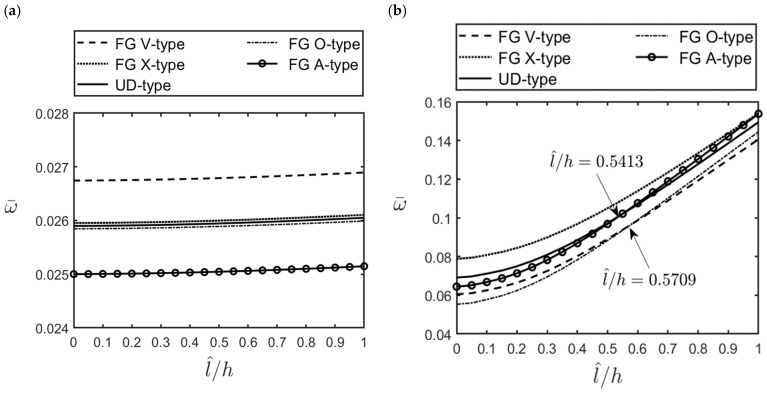
Variations in the lowest dimensionless natural frequencies of an FG-GPLRC thick cylindrical microshell with the ratio of $\hat{l}/h$ for different GPL distribution patterns and different wave number pairs: (**a**) $\left(\hat{m},\hat{n}\right)$ = (1, 1), (**b**) $\left(\hat{m},\hat{n}\right)$ = (1, 2), for which *R*/*h* = 5, *L*/*R* = 10, and $W_{GPL}^{*}=0.01$.

**Figure 6 materials-16-02363-f006:**
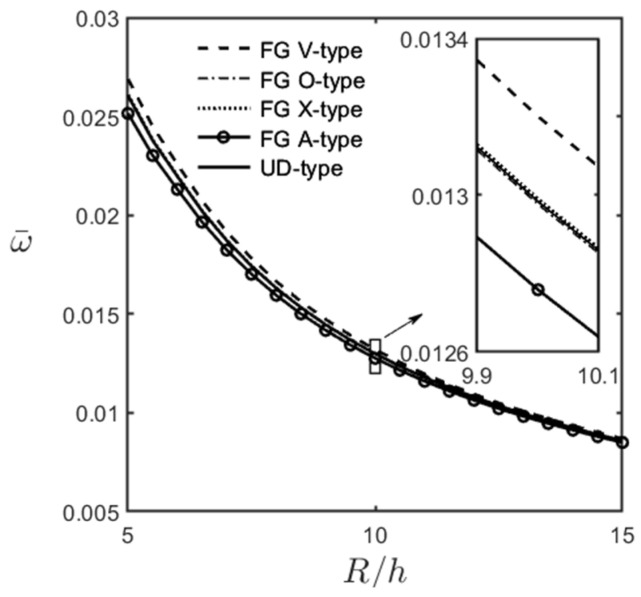
Variations in the dimensionless fundamental frequencies of an FG-GPLRC cylindrical microshell with the ratio of *R*/*h* for different GPL distribution patterns, for which $\left(\hat{m},\hat{n}\right)$ = (1, 1), *L*/*R* = 10, $\hat{l}/h=1.0$, and $W_{GPL}^{*}=0.01$.

**Figure 7 materials-16-02363-f007:**
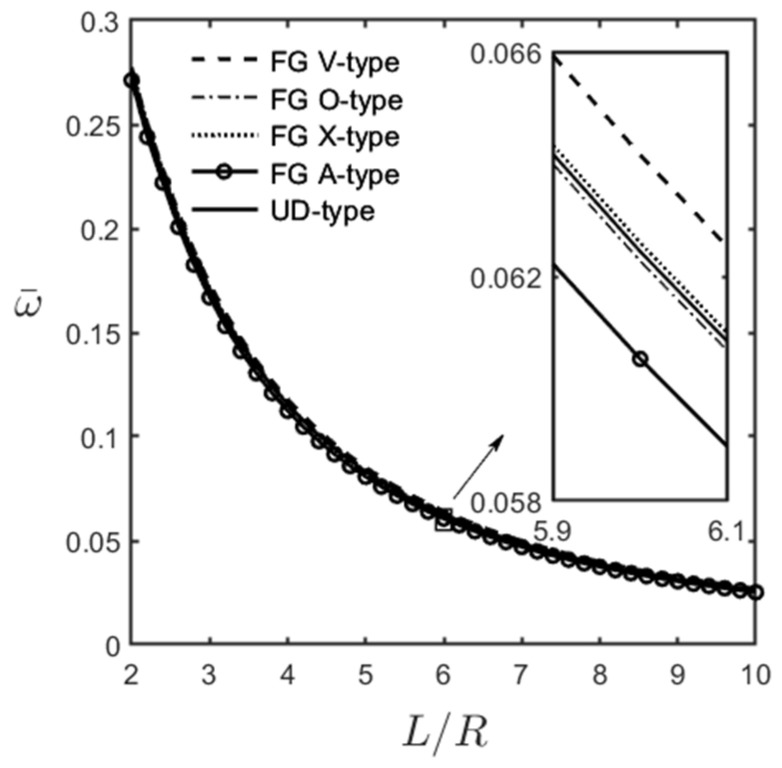
Variations in the dimensionless fundamental frequencies of an FG-GPLRC cylindrical microshell with the ratio of *L*/*R* for different GPL distribution patterns, for which $\left(\hat{m},\hat{n}\right)$ = (1, 1), *R*/*h* = 5, $\hat{l}/h=1.0$, and $W_{GPL}^{*}=0.01$.

**Figure 8 materials-16-02363-f008:**
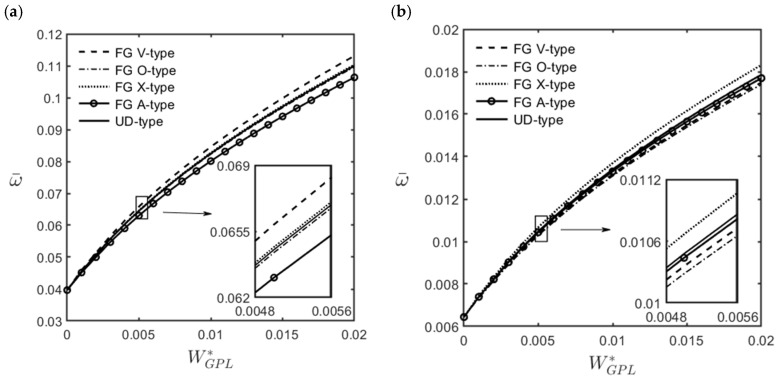
Variations in the dimensionless fundamental frequencies of an FG-GPLRC cylindrical microshell with the weight fraction of GPL for different GPL distribution patterns, for which $\hat{l}/h=1.0$, and (**a**) *R*/*h* = 5, *L*/*R* = 5, and $\left(\hat{m},\hat{n}\right)$ = (1, 1), (**b**) *R*/*h* = 20, *L*/*R* = 5, and $\left(\hat{m},\hat{n}\right)$ = (1, 2).

**Figure 9 materials-16-02363-f009:**
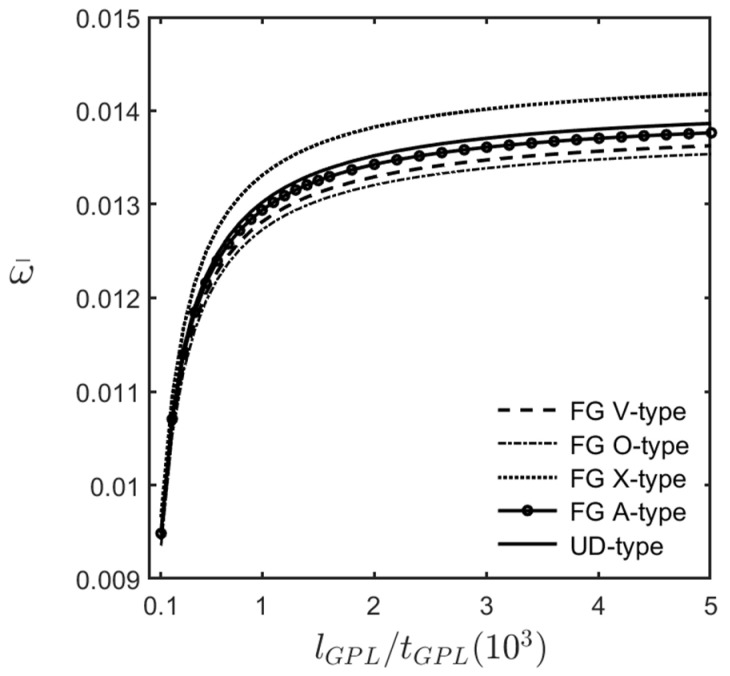
Variations in the dimensionless fundamental frequencies of an FG-GPLRC cylindrical microshell with the length–to–thickness ratio of GPL for different GPL distribution patterns, for which $\left(\hat{m},\hat{n}\right)$ = (1, 2), $\hat{l}/h=1.0$, *R*/*h* = 20, *L*/*R* = 5, and $W_{GPL}^{*}=0.01$.

**Table 1 materials-16-02363-t001:** Convergence and validation studies for the lowest frequency parameter solutions of simply-supported, homogeneous isotropic SWCNTs obtained using two-node CCST-based FLMs and the MCST-based LCST, where *h*/*R* = 0.05.

$\left(\hat{m},\hat{n}\right)$	$\hat{l}/h$	Zeighampour and Beni [46]	Two-Node CCST-Based FLMs							
			Lagrangian *C*^0^ FLMs			Hermitian *C*^1^ FLMs		Hermitian *C*^2^ FLMs		
			*n_l_* = 1	*n_l_* = 2	*n_l_* = 4	*n_l_* = 1	*n_l_* = 2	*n_l_* = 1	*n_l_* = 2	Relative Errors
(1, 1)	0	0.19588	0.195441	0.195435	0.195434	0.195434	0.195434	0.195434	0.195434	0.2%
	1	0.21109	0.196435	0.196430	0.196428	0.196066	0.196065	0.196065	0.196065	7.7%
(2, 2)	0	0.26245	0.260008	0.258832	0.258534	0.258434	0.258434	0.258434	0.258434	1.6%
	1	0.39538	0.360147	0.359365	0.359167	0.357885	0.357883	0.357883	0.357883	10.5%
(3, 3)	0	0.32242	0.322392	0.315408	0.313598	0.312990	0.312990	0.312990	0.312990	3.0%
	1	0.76443	0.732952	0.730358	0.729696	0.727972	0.727969	0.727969	0.727969	5.0%
(4, 4)	0	0.43010	0.436538	0.418090	0.413173	0.411505	0.411505	0.411505	0.411505	3.0%
	1	1.34110	1.296770	1.292101	1.290895	1.288570	1.288566	1.288566	1.288566	4.1%
(5, 5)	0	0.60367	0.610821	0.577106	0.567870	0.564713	0.564712	0.564712	0.564712	6.9%
	1	2.20417	2.016211	2.009530	2.007785	2.003963	2.003955	2.003954	2.003954	10.0%

Relative errors = 100% × (Zeighampour and Beni’s solutions − Hermitian *C*^2^ FEM solutions)/Hermitian *C*^2^ FEM solutions.

**Table 2 materials-16-02363-t002:** Convergence and validation studies for the natural frequencies (Hz) of simply-supported FG cylindrical macroshells obtained using the two-node and three-node Hermitian *C*^2^ FCLMs, with different values of wave number pairs $\left(\hat{m},\hat{n}\right)$ and $\kappa_{p}$.

$\kappa_{p}$	Theories	$\left(\hat{m},\hat{n}\right)$										
		(1, 0)	(1, 1)	(1, 2)	(1, 3)	(1, 4)	(1, 5)	(1, 6)	(1, 7)	(1, 8)	(1, 9)	(1, 10)
0.5	Two-node Hermitian *C*^2^ ($n_{l}=2$)	76.4537	13.3336	4.5196	4.1850	7.0831	11.3125	16.5600	22.7794	29.9612	38.1026	47.2026
	Two-node Hermitian *C*^2^ ($n_{l}=4$)	76.4536	13.3336	4.5198	4.1846	7.0831	11.3125	16.5599	22.7795	29.9611	38.1026	47.2025
	Three-node Hermitian *C*^2^ ($n_{l}=2$)	76.4537	13.3337	4.5197	4.1847	7.0832	11.3126	16.5600	22.7794	29.9612	38.1026	47.2026
	Three-node Hermitian *C*^2^ ($n_{l}=4$)	76.4536	13.3337	4.5197	4.1844	7.0830	11.3130	16.5602	22.7793	29.9613	38.1026	47.2026
	Liu et al. [49]	NA	13.341	4.5295	4.1830	7.0913	11.328	16.583	22.811	30.002	38.155	47.267
	Loy et al. [48]	NA	13.321	4.5168	4.1911	7.0972	11.336	16.594	22.826	30.023	38.181	47.301
1	Two-node Hermitian *C*^2^ ($n_{l}=2$)	75.8588	13.2236	4.4827	4.1508	7.0243	11.2182	16.4216	22.5888	29.7104	37.7835	46.8073
	Two-node Hermitian *C*^2^ ($n_{l}=4$)	75.8588	13.2236	4.4829	4.1507	7.0244	11.2182	16.4215	22.5888	29.7104	37.7836	46.8073
	Three-node Hermitian *C*^2^ ($n_{l}=2$)	75.8588	13.2236	4.4826	4.1508	7.0242	11.2182	16.4215	22.5888	29.7104	37.7836	46.8073
	Three-node Hermitian *C*^2^ ($n_{l}=4$)	75.8588	13.2236	4.4829	4.1512	7.0246	11.2184	16.4215	22.5888	29.7105	37.7836	46.8072
	Liu et al. [49]	NA	13.242	4.480	4.1459	7.0325	11.229	16.438	22.612	29.741	37.822	46.855
	Loy et al. [48]	NA	13.211	4.480	4.1569	7.0384	11.241	16.455	22.635	29.771	37.862	46.905
30	Two-node Hermitian *C*^2^ ($n_{l}=2$)	74.2516	12.9261	4.3793	4.0517	6.8590	10.9559	16.0387	22.0628	29.0189	36.9045	45.7185
	Two-node Hermitian *C*^2^ ($n_{l}=4$)	74.2516	12.9261	4.3790	4.0518	6.8589	10.9560	16.0387	22.0628	29.0189	36.9045	45.7185
	Three-node Hermitian *C*^2^ ($n_{l}=2$)	74.2516	12.9262	4.3793	4.0515	6.8590	10.9559	16.0386	22.0628	29.0189	36.9045	45.7185
	Three-node Hermitian *C*^2^ ($n_{l}=4$)	74.2516	12.9259	4.3791	4.0526	6.8590	10.9561	16.0388	22.0629	29.0190	36.9045	45.7185
	Liu et al. [49]	NA	12.945	4.3810	4.0469	6.8561	10.949	16.029	22.049	29.001	36.882	45.691
	Loy et al. [48]	NA	12.914	4.3765	4.0576	6.8726	10.978	16.071	22.108	29.078	36.981	45.813

**Table 3 materials-16-02363-t003:** Comparison study for the first three lowest frequency parameters ${\bar{\omega }}_{k}$ of laminated GPLRC cylindrical macroshells ($\hat{l}/h=0$) and microshells ($\hat{l}/h\neq 0$) obtained using the CCST-based three-node Hermitian *C*^2^ FCLM, for which $\hat{m}$ = 2, 5, and 10; $n_{l}$ = 20; and ${\bar{\omega }}_{k}=\omega_{k}\left[R-\left(h/2\right)\right]\sqrt{\rho_{epoxy}/E_{epoxy}}$.

$\hat{m}$	GPL Distribution Patterns	${\bar{\omega }}_{1}$($\hat{n}$)					${\bar{\omega }}_{2}$($\hat{n}$)					${\bar{\omega }}_{3}$($\hat{n}$)				
		$\hat{l}/h=0$ Liu et al. [49]	$\hat{l}/h=0$	$\hat{l}/h=0.2$	$\hat{l}/h=0.4$	$\hat{l}/h=1.0$	$\hat{l}/h=0$ Liu et al. [49]	$\hat{l}/h=0$	$\hat{l}/h=0.2$	$\hat{l}/h=0.4$	$\hat{l}/h=1.0$	$\hat{l}/h=0$ Liu et al. [49]	$\hat{l}/h=0$	$\hat{l}/h=0.2$	$\hat{l}/h=0.4$	$\hat{l}/h=1.0$
2	Epoxy	0.9659	0.9659 (1)	1.0320 (1)	1.1644 (1)	1.3349 (1)	0.9996	0.9996 (0)	1.0707 (0)	1.2431 (0)	1.3442 (0)	1.0161	1.0161 (2)	1.0999 (2)	1.2576 (2)	1.4874 (2)
	UD	2.3674	2.3674 (1)	2.5300 (1)	2.8555 (1)	3.2745 (0)	2.4499	2.4499 (0)	2.6247 (0)	3.0482 (0)	3.2972 (1)	2.4904	2.4904 (2)	2.6964 (2)	3.0842 (2)	3.6485 (2)
	FG V-type	2.2362	2.2359 (1)	2.4412 (1)	2.8687 (1)	3.4376 (1)	2.2518	2.2516 (0)	2.4518 (0)	2.9132 (0)	3.4590 (0)	2.3753	2.3750 (2)	2.6221 (2)	3.0965 (2)	3.7089 (2)
	FG A-type	2.2517	2.2515 (1)	2.3613 (1)	2.5086 (1)	2.5664 (0)	2.3636	2.3633 (2)	2.5189 (0)	2.5403 (0)	2.6216 (1)	2.4659	2.4653 (0)	2.5321 (2)	2.7482 (2)	2.9088 (2)
	FG X-type	2.3030	2.3028 (1)	2.5173 (1)	2.8614 (1)	3.4279 (0)	2.3663	2.3663 (2)	2.6274 (0)	3.0562 (0)	3.4285 (1)	2.4109	2.4107 (0)	2.6586 (2)	3.0813 (2)	3.7938 (2)
	FG O-type	2.2171	2.2170 (1)	2.4044 (1)	2.7498 (1)	2.9874 (0)	2.2915	2.2913 (0)	2.4900 (0)	2.9528 (0)	3.0260 (1)	2.3348	2.3347 (2)	2.5687 (2)	2.9735 (2)	3.3336 (2)
5	Epoxy	2.6997	2.6997 (0)	3.2633 (1)	3.6532 (0)	3.9184 (0)	2.7156	2.7156 (1)	3.2779 (0)	3.6761 (1)	3.9446 (1)	2.7680	2.7680 (2)	3.3038 (2)	3.7496 (2)	4.0230 (2)
	UD	6.6185	6.6185 (0)	8.0037 (1)	8.9610 (0)	9.6115 (0)	6.6576	6.6576 (1)	8.0390 (0)	9.0173 (1)	9.6760 (1)	6.7861	6.7861 (2)	8.1031 (2)	9.1975 (2)	9.8682 (2)
	FG V-type	6.2760	6.2172 (0)	6.8898 (0)	7.1421 (0)	7.3011 (0)	6.2435	6.2437 (1)	6.9063 (1)	7.1761 (1)	7.3372 (1)	6.3383	6.3387 (2)	6.9986 (2)	7.2997 (2)	7.4661 (2)
	FG A-type	5.4878	5.4895 (0)	5.7000 (0)	5.8302 (0)	5.9099 (0)	5.5120	5.5137 (1)	5.7298 (1)	5.8620 (1)	5.9422 (1)	5.5865	5.5882 (2)	5.8192 (2)	5.9563 (2)	6.0376 (2)
	FG X-type	6.1009	6.1012 (0)	7.6554 (1)	9.1384 (0)	11.2431 (0)	6.1354	6.1357 (1)	7.7157 (0)	9.1948 (1)	11.3290 (1)	6.2496	6.2499 (2)	7.7429 (2)	9.3989 (2)	11.5879 (2)
	FG O-type	NA	6.5361 (0)	6.9735 (0)	7.1062 (0)	7.1863 (0)	NA	6.5658 (1)	7.0069 (1)	7.1451 (1)	7.2257 (1)	NA	6.6742 (2)	7.1128 (2)	7.2609 (2)	7.3422 (2)
10	Epoxy	5.6503	5.6503 (0)	7.5607 (0)	8.0905 (0)	8.2791 (0)	5.6603	5.6603 (1)	7.5733 (1)	8.1027 (1)	8.2911 (1)	5.6897	5.6897 (2)	7.6109 (2)	8.1393 (2)	8.3270 (2)
	UD	13.8540	13.8540 (0)	18.5459 (0)	19.8455 (0)	20.3082 (0)	13.8785	13.8785 (1)	18.5767 (1)	19.8755 (1)	20.3376 (1)	13.9504	13.9504 (2)	18.6690 (2)	19.9652 (2)	20.4257 (2)
	FG V-type	10.3491	10.3531 (0)	11.7683 (0)	12.1279 (0)	12.2624 (0)	10.3520	10.3560 (1)	11.8016 (1)	12.1623 (1)	12.2969 (1)	10.3782	10.3822 (2)	11.9024 (2)	12.2663 (2)	12.4010 (2)
	FG A-type	9.6421	9.6458 (0)	10.6379 (0)	10.8929 (0)	10.9890 (0)	9.6527	9.6563 (1)	10.6519 (1)	10.9071 (1)	11.0031 (1)	9.6843	9.6880 (2)	10.6937 (2)	10.9494 (2)	11.0454 (2)
	FG X-type	12.0604	12.0645 (0)	16.9969 (0)	23.9794 (0)	24.0569 (0)	12.0798	12.0839 (1)	17.0399 (1)	24.0224 (1)	24.0996 (1)	12.1380	12.1422 (2)	17.1694 (2)	24.1506 (2)	24.2267 (2)
	FG O-type	11.7123	11.7164 (0)	12.6699 (0)	12.8855 (0)	12.9659 (0)	11.7251	11.7291 (1)	12.6857 (1)	12.9014 (1)	12.9818 (1)	11.7632	11.7673 (2)	12.7328 (2)	12.9490 (2)	13.0293 (2)

**Table 4 materials-16-02363-t004:** Solutions for the lowest frequency parameters $\bar{\omega }$ of FG-GPLRC cylindrical microshells obtained using three-node Hermitian *C*^2^ FEM, with different wave number values, $n_{l}=20$, L/R=10 and $\bar{\omega }=\omega h\sqrt{\rho_{epoxy}/E_{epoxy}}$.

$\left(\hat{m},\hat{n}\right)$	*R*/*h*	Types	$\hat{l}/h$ (*l*/*h*)					
			0 (0)	0.2 (0.1)	0.4 (0.2)	0.6 (0.3)	0.8 (0.4)	1.0 (0.5)
(1, 1)	5	FG V-type	0.03156	0.03157	0.03160	0.03163	0.03168	0.03174
		FG X-type	0.03056	0.03057	0.03059	0.03063	0.03068	0.03074
		UD	0.03049	0.03050	0.03052	0.03056	0.03061	0.03067
		FG O-type	0.03042	0.03043	0.03045	0.03049	0.03053	0.03059
		FG A-type	0.02934	0.02935	0.02937	0.02941	0.02945	0.02951
(1, 1)	10	FG V-type	0.01546	0.01546	0.01546	0.01547	0.01547	0.01548
		FG X-type	0.01519	0.01519	0.01520	0.01520	0.01521	0.01521
		UD	0.01518	0.01518	0.01519	0.01519	0.01520	0.01521
		FG O-type	0.01517	0.01518	0.01518	0.01518	0.01519	0.01520
		FG A-type	0.01490	0.01490	0.01490	0.01491	0.01491	0.01492
(1, 1)	20	FG V-type	0.00765	0.00765	0.00765	0.00765	0.00765	0.00766
		FG X-type	0.00758	0.00759	0.00759	0.00759	0.00759	0.00759
		UD	0.00758	0.00758	0.00758	0.00758	0.00759	0.00759
		FG O-type	0.00758	0.00758	0.00758	0.00758	0.00758	0.00759
		FG A-type	0.00751	0.00751	0.00751	0.00751	0.00752	0.00752
(1, 2)	10	FG V-type	0.01845	0.02011	0.02441	0.03021	0.03678	0.04372
		FG X-type	0.02468	0.02606	0.02970	0.03487	0.04096	0.04758
		UD	0.02115	0.02269	0.02677	0.03241	0.03890	0.04584
		FG O-type	0.01661	0.01854	0.02337	0.02969	0.03670	0.04402
		FG A-type	0.01906	0.02086	0.02548	0.03167	0.03865	0.04601
(2, 2)	10	FG V-type	0.02732	0.02867	0.03237	0.03771	0.04404	0.05095
		FG X-type	0.03234	0.03359	0.03697	0.04192	0.04792	0.05459
		UD	0.02935	0.03067	0.03431	0.03960	0.04593	0.05288
		FG O-type	0.02578	0.02728	0.03134	0.03708	0.04382	0.05109
		FG A-type	0.02752	0.02900	0.03302	0.03877	0.04554	0.05289
(1, 3)	10	FG V-type	0.04901	0.05376	0.06591	0.08210	0.10023	0.11922
		FG X-type	0.06623	0.07034	0.08066	0.09502	0.11177	0.12980
		UD	0.05686	0.06124	0.07271	0.08842	0.10633	0.12529
		FG O-type	0.04392	0.04949	0.06325	0.08099	0.10042	0.12051
		FG A-type	0.05113	0.05627	0.06937	0.08673	0.10607	0.12625
(2, 3)	10	FG V-type	0.05140	0.05624	0.06868	0.08531	0.10394	0.12345
		FG X-type	0.06883	0.07306	0.08365	0.09840	0.11561	0.13412
		UD	0.05933	0.06380	0.07555	0.09167	0.11007	0.12953
		FG O-type	0.04622	0.05188	0.06592	0.08409	0.10404	0.12466
		FG A-type	0.05347	0.05870	0.07208	0.08986	0.10970	0.13040
(3, 3)	10	FG V-type	0.05705	0.06194	0.07464	0.09178	0.11111	0.13142
		FG X-type	0.07450	0.07885	0.08979	0.10507	0.12295	0.14220
		UD	0.06492	0.06947	0.08152	0.09817	0.11725	0.13749
		FG O-type	0.05185	0.05749	0.07173	0.09040	0.11105	0.13248
		FG A-type	0.05896	0.06424	0.07786	0.09616	0.11670	0.13819

## Data Availability

The processed data required to reproduce these findings are available to download from [https://drive.google.com/drive/folders/1lnzd0IlWRXM51d550tJhh5px5v9IsO54?usp=share_link] (accessed on 12 March 2023).

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
