# Peer review of "A Size-Dependent Finite Element Method for the 3D Free Vibration Analysis of Functionally Graded Graphene Platelets-Reinforced Composite Cylindrical Microshells Based on the Consistent Couple Stress Theory"

_materials, 2023, doi:10.3390/ma16062363_

Round 1

Reviewer 1 Report

A size dependent FEM for the 3D free vibration analysis of FG cylindrical microshells have been reported with an extensive parametric study. The authors have carried out excellent work by extending the theory developed by them for the study on microplates to microshells within the framework of (CCST).

To enhance this work further the following comments may be considered

Having developed a 3D solution, it’s better that any of the available 2D theories could have been compared to bring out the importance of 3D effects, thus justifying the importance of this 3D solution.

Having developed the FEA solution, the work could have included the parametric study for other different boundary conditions where these 3D effects are more predominant.

The authors have done a meticulous work, it's perfect, except a few typo and formatting error as follows for corrections

Line 325 a space between the words “node”  and “Hermitian”

Line 566 “moderately thick” can be used for “moderate thick”

In References: Spacing between the reference nos and the references need to be uniform, particularly please check Ref no 22,35,42,44,45 & 48 

Reviewer 2 Report

1-     Please arrange the Abstract section more precisely. It is necessary to mention different types of GPLs distribution patterns in this section. In the first line of the Abstract section, change the statement “we develop a size‐ dependent finite element method” to the passive form. Also, the mentioned sentence is too long. I think the “in‐surface and out‐of‐surface” expressions should be replaced by “in-plane and out-of-plane” ones. The utilized models of shells have not been explained in the Abstract section.

2-     Authors are expected to mark their most important novelty in the Abstract section. It is obvious that the reinforced cylindrical shell has a higher natural frequency in comparison to the isotropic homogenous one. Quantitative results have not been covered in this section. 

3-     The grammar of the manuscript is poor and an exact modification is required. Authors should check the use of punctuation marks in the manuscript. For instance:

·        The complete form and abbreviation of “GPLs” have been repeated (see lines 13  and 38).

·        “precision industry” (see line 41)

·        The complete form and abbreviation “GPLRC” have been repeated (see lines 44 and 13).

·        “size‐dependency effect”(see lines 46,49)

·        “… microstructures when their dimensions reduce to the microscales and even the nanoscales” (see line 48)

·        “Because of Eringen’s criticism [19] for the indeterminacy problems of the CST…”

(see line 55)

·        the material length scale coefficient required for analyzing elastic isotropic solids was reduced from two to one (see line 59)

·        The reference of line 63 has been missed.

·        Hamilton principle (Hamilton’s principle)

·        Bolotin method

·        The results obtained by Wu and Hsu were shown

·        by assigning the material length scale parameter to the value of zero à by ignoring the material length scale parameter

·        they are expressed following Yang et al. [43] asàthey are expressed as [43]

4-     In the Introduction section in the statement “The results showed that the overall stiffness of the nanoshell is greater in the MCST than in the LCST”, authors have compared a constitutive theory (MCST) with a shell theory (LCST). Mentioned comparison is not a scientific one.

5-     Replace the word “variation” with “distribution” in GPLs distribution patterns (Eqs. (10a)-(10e).

6-     In my opinion section 3 is not readable because the Eq. of and some tensors have been ignored.

7-     Authors should clarify the advantages and disadvantages of CST, MST, and CCST based on their relations in the introduction

8-     The CCST equations have been mentioned briefly. So, the paper is not readable for researchers.

9-     Following papers are advised to improve introduction section

https://doi.org/10.1007/s42417-022-00729-z, https://doi.org/10.1038/s41598-022-09043-w,

Round 2

Reviewer 2 Report

The requested corrections have been completed in the current version by the Authors. I find  this paper suitable for publishing.